# BOSS: Diversity-Difficulty Balanced One-Shot Subset Selection for Data-Efficient Deep Learning

## Abstract

Subset or core-set selection offers a data-efficient way for training deep learning models by identifying important data samples so that the model can be trained using the selected subset with similar performance as trained on the full set. However, most existing methods tend to choose either diverse or difficult data samples, which is likely to form a suboptimal subset, leading to a model with compromised generalization performance. One key limitation is due to the misalignment with the underlying goal of subset selection as an optimal subset should faithfully represent the joint data distribution that is comprised of both feature and label information. To this end, we propose to conduct diversity-difficulty Balanced One-shot Subset Selection (BOSS), aiming to construct an optimal subset for data-efficient deep learning. Samples are selected into the subset so that a novel balanced core-set loss bound is minimized, which theoretically justifies the need to simultaneously consider both diversity and difficulty to form an optimal subset. The loss bound also unveils the key relationship between the type of data samples to be included in the subset and the subset size. This further inspires the design of an expressive importance function to optimally balance diversity and difficulty depending on the subset size. The proposed approach is inspired by a theoretical loss bound analysis and utilizes a fine-grained importance control mechanism. A comprehensive experimental study is conducted on both synthetic and real datasets to justify the important theoretical properties and demonstrate the superior performance of BOSS as compared with the competitive baselines.

## 1 Introduction

Deep learning has enjoyed a great popularity in a wide range of domains, including natural language processing (Brown et al., 2020; Liu et al., 2019), computer vision (Ramesh et al., 2021; Dosovitskiy et al., 2021; Tan & Le, 2019; Chen et al., 2020) and more. However, the success comes at a cost of a large amount of data and increased resource consumption. Such resources include computational cost, training time, energy usage, financial burden, and carbon emission (Schwartz et al., 2020; Strubell et al., 2019). The resource consumption is usually proportional to the amount of data used for training (Hestness et al., 2017) and at the same time, using large amounts of data is not always feasible because of constraints such as labeling cost (Settles, 2009), memory limitation (Shin et al., 2017), and sparse computing resources (Konečnỳ et al., 2016) (*e.g.,* on mobile or edge devices).

Subset or core-set selection aims to find candidate data points from a large pool of data such that the model trained on the subset has comparable performance to that of the model trained on the full set (Feldman, 2020). This will in turn help decrease the resources consumed by training on large amounts of data. Intuitively, subsets can be chosen dynamically during each training epoch (Mirzasoleiman et al., 2020; Killamsetty et al., 2021b;a; Pooladzandi et al., 2022). However, the selection algorithm is usually time-consuming and can significantly increase the overall training duration (Shin et al., 2023). Such a process also requires a forward pass through the entire dataset each time a subset is chosen, which incurs a high cost for a large dataset. On the other hand, *one-shot subset selection* only picks the subset once before the training starts and uses that subset for the entire training process (Zheng et al., 2023; Paul et al., 2021; Feldman & Zhang, 2020; Sorscher et al., 2022).While it may still be essential to initialize a model using the full dataset for few epochs

to obtain the training dynamics employed for subset selection, one-shot subset selection offers the advantage that the time required for this selection is accounted for only once. One-shot subset selection can also serve for the scenarios such as continual learning (Nguyen et al., 2018).

Subset selection have been used for classical problems such as regression (Madigan et al., 2002), classification (Tsang et al., 2005), and clustering (Har-Peled & Kushal, 2005). Recent works have started exploring applications of core-set selection for data-efficient deep learning (Guo et al., 2022; Wan et al., 2022; Killamsetty et al., 2021c). Two categories of methods have been explored to incorporate the most important examples into the selected subset, including 1) diversity based, which selects a diverse set of samples to cover the entire feature (or gradient) space (Mirzasoleiman et al., 2020; Killamsetty et al., 2021a;b; Pooladzandi et al., 2022; Shin et al., 2023; Welling, 2009; Agarwal et al., 2020; Sener & Savarese, 2018), and 2) difficulty-based, which selects the most difficult samples to best characterize the decision boundary (Toneva et al., 2019; Feldman & Zhang, 2020; Paul et al., 2021; Sorscher et al., 2022). More specifically, diversity-based methods leverage the facility location objective (Farahani & Hekmatfar, 2009) to select the optimal subset such that the distance between the subset and full set in the feature (Sener & Savarese, 2018; Welling, 2009; Agarwal et al., 2020) or gradient space (Mirzasoleiman et al., 2020; Killamsetty et al., 2021a; Pooladzandi et al., 2022; Shin et al., 2023) is minimized. In contrast, difficulty-based methods (Toneva et al., 2019; Paul et al., 2021) score examples based on a difficulty metric where a high score corresponds to a higher difficulty. Paul et al. (2021) shows that by removing the easier examples, a large portion (*i.e.,* 25%–50%) of a full dataset can be pruned without obviously compromising the model's generalization performance.

However, the existing methods as described above are fundamentally limited as they are inadequate to select the optimal subset of samples. This is due to the fact that the selection criterion does not align with the ultimate goal of subset selection, which is to represent a joint distribution $P(\mathbf{x}, \mathbf{y})$ (as instantiated by a full set) using a small subset of data samples. Consequently, solely relying on the feature (*i.e.,* $\mathbf{x}$) or the label side (*i.e.,* $\mathbf{y}$) will lead to a suboptimal selection result. There have been some recent efforts, aiming to improve the existing one-shot core-set selection methods (Zheng et al., 2023; Xia et al., 2023), where they introduce difficulty metrics or new sampling strategy using existing difficult metrics.

For example, coverage-centric core-set selection (CCS) (Zheng et al., 2023) considers various levels of difficulty when choosing samples to form the subset, which helps to improve the quality of the subset. Nevertheless, a principled way is still lacking to properly balance diversity and difficulty for choosing a subset that can faithfully represent the underlying joint distribution. As Figure 1 (a) shows, the subset chosen by CCS (as highlighted

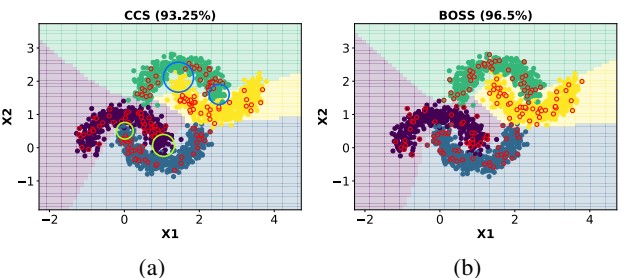

Figure 1: CCS (a) Vs. BOSS (b): Decision boundary learned using the selected subset shown in red circles by CCS and BOSS, where the subset is 10% of the full set.

in red) misses some critical regions (*e.g.,* as annotated in the figure) in the full set, leading to a suboptimal subset with a lower generalization performance (due to a less accurate decision boundary).

To address the key limitations of existing approaches, we propose to perform diversity-difficulty **B**alanced **O**ne-shot **S**ubset **S**election (BOSS), aiming to construct an optimal subset to achieve data-efficient deep learning. BOSS performs subset selection guided by a balanced core-set loss bound that reveals an important trade-off between feature similarity (*i.e.,* diversity) and label variability (*i.e.,* difficulty). In particular, the balanced loss bound is comprised of two key components as a natural result of the joint impact from the feature and label sides, respectively. This theoretical result further confirms the need to properly model the joint data distribution in subset selection as solely relying on the feature or label sides will result in a significantly loose loss bound that will compromise the learning process. Furthermore, the novel loss bound also uncovers important relationship between the type of data samples to be selected (*i.e.,* diverse or difficult) and the size of the subset (as determined by the available computing budget). In particular, for a small subset size, the loss introduced by the feature similarity will dominate the bound, which will direct subset selection to

choose representative samples to avoid using dissimilar samples in the subset to represent samples in the full set. As the size increases, the large label variability from certain (*i.e.,* difficult) regions in the joint distribution will contribute more significantly to the overall loss bound. This will force the selection of samples from these regions so that the decision-boundary can be further refined to reduce the label loss. This key theoretical insight suggests that when integrating diversity and difficulty for subset selection, subset size plays a crucial role. To this end, we design an expressive importance function that can properly balance diversity and difficulty depending on the subset size.

Figure 1 (b) visualizes the subset chosen by the proposed BOSS method. As compared with CCS, BOSS adequately covers the entire feature space while attending to all critical regions, which ensures that an accurate decision boundary can be learned from the chosen subset with a much improved prediction performance than CCS. Our main contribution is threefold: (1) a novel balanced core-set loss bound which not only justifies the necessity of simultaneously considering both diversity and difficulty for subset selection but also unveils the key relationship between the type of data samples to be included in the subset and the subset size, (2) design of an expressive importance function to optimally balance diversity and difficulty for subset selection given the subset size, and (3) a comprehensive evaluation using both synthetic and real-world data to verify the key theoretical results and empirical performance of the proposed method.

## 2 RELATED WORK

### 2.1 DIVERSITY-BASED SUBSET SELECTION

**Gradient-based subset selection (GB-SS).** GB-SS aims to find a subset such that the difference between the sum of the gradients of the full set and the weighted sum of the gradients of the subset is minimized. As a representative GB-SS method, CRAIG (Mirzasoleiman et al., 2020) shows the gain for convex optimization or simple classification tasks but becomes less competitive for complex deep learning models and difficult learning tasks. GradMatch (Killamsetty et al., 2021a) improves on CRAIG by regularizing the weight values such that large weight values are penalized while selecting the subset. Adacore (Pooladzandi et al., 2022) leverages a Hessian pre-conditioned gradient to capture the curvature information of gradient and exponential moving average of gradients to smooth out the local gradient information. In addition to minimizing the gradient difference, LCMAT (Shin et al., 2023) also minimizes the difference of maximum eigenvalue obtained from the inverse of Hessian of full set and subset in order to capture the curvature information of the loss landscape. Although these methods improve the performance, the calculation of inverse Hessian approximation is time-consuming and computationally expensive (Pearlmutter, 1994).

**Feature-based subset selection (FB-SS).** FB-SS aims to find a representative subset in the feature space. $K$-center (Farahani & Hekmatfar, 2009) is a mini-max facility location problem where the subset is selected such that the maximum distance between a point in the original dataset closest to the chosen center is minimized. Herding (Welling, 2009) selects the subset such that the distance between the centroid of the full set and the subset is minimized. The centroid is found using the feature of the input. Contextual Diversity (Agarwal et al., 2020) improves the visual diversity in the feature space and uses KL divergence for calculating the pairwise distance. Although these methods leverage input features to select the subset, they do not consider the sample difficulty. However, the difficulty level of the samples is important because even if two samples are close in the feature space, they can have distinct difficulty scores, especially for those close to the decision boundary.

### 2.2 DIFFICULTY-BASED SUBSET SELECTION

Difficulty-based subset selection scores each example based on some difficulty metric that measures how difficult it is to learn the sample or how much impact the sample has on the generalization. Toneva et al. (2019) count the number of times an example is learned and then forgotten to identify which examples are difficult. The number of times an example is forgotten is denoted as the forgetting score. The larger the difficulty, the higher the score. To select the subset, the easier samples are discarded and only the difficult samples are used. Although the forgetting score gives a good estimate of the difficulty of a sample, computing the scores requires training the model on the full dataset multiple times to get a reliable score for the sample. Paul et al. (2021) introduce the EL2N score which stands for $L_2$ norm of a prediction error. Unlike forgetting scores, EL2N can be calculated early on during the training such that the time to find the subset is significantly lower. Sorscher et al. (2022) compare the EL2N score with other scores such as the influence score (Feldman &

Zhang, 2020) to select the subset. The influence score of a sample is the measure of how much the generalization performance of a model suffers if that sample is removed from the training dataset. Samples with high influence scores are deemed more difficult. However, this method is also computationally expensive because it needs to train the model multiple times on the full dataset. Although difficulty-based methods prove to be effective for larger subset sizes, they tend to choose suboptimal solutions when the subset size is small. Our theoretical results reveal the key underlying reason for this behavior. One very recent work (Xia et al., 2023) defines a new difficulty metric based on the distance of each example with the center of the related class such that we can select the samples with smaller distances to their class center. However, it ignores the diversity in the feature space. Another recent work (Zheng et al., 2023) develops a new sampling method that can utilize different difficulty scores to achieve better performance compared to only selecting the most difficult samples. It selects samples randomly among different strata of difficulty scores and allocates an equal budget among the strata. Our theoretical results show that the diversity and difficult components need to be carefully balanced to avoid a loose loss bound that can misguide the subset selection process.

## 3 METHODOLOGY

Consider a deep learning model with parameters $\boldsymbol{\theta}$ and a training dataset $\mathcal{V} = \{\mathbf{x}_i, \mathbf{y}_i\}_{i=1}^{|\mathcal{V}|}$ from which we want to select a subset $\mathcal{S} \subseteq \mathcal{V}$. We use one-hot vectors for the labels $\mathbf{y}$. The training objective is to find the set of parameters $\boldsymbol{\theta}$ that gives us the lowest training error $l = \frac{1}{|\mathcal{T}|} \sum_{n=1}^{|\mathcal{T}|} l_n(\boldsymbol{\eta}(\mathbf{x}_n), \mathbf{y}_n; \boldsymbol{\theta})$, where $\mathcal{T}$ could be either the full set or the subset and $\boldsymbol{\eta}(\mathbf{x}_n) = (\eta^{(1)}, ...\eta^{(K)})^\top$ is the model prediction for $\mathbf{x}_n$ given $\boldsymbol{\theta}$. To obtain the optimal model that can be trained over $\mathcal{S}$, we first train a model for a few epochs on the full dataset (i.e., $\mathcal{T} = \mathcal{V}$) to select the subset $\mathcal{S}$ from $\mathcal{V}$ using the model information. A newly initialized model $\boldsymbol{\theta}_{\mathcal{S}}$ is then trained on the subset ($\mathcal{T} = \mathcal{S}$). The size $|\mathcal{S}|$ is limited by the amount of budget or resources available. We want to find a subset such that the model trained on the subset has a comparable generalization capability to that of the model trained on the full set.

### 3.1 THE BALANCED CORE-SET LOSS BOUND

Our goal is to find the optimal subset that generalizes similarly to the model trained on the full set. Following the core-set based formulation (Sener & Savarese, 2018), the true generalization loss of the model $\theta_{\mathcal{S}}$ is closely related to the full set loss:

$$\mathbb{E}_{\mathbf{x},\mathbf{y}}\left[l(\boldsymbol{\eta}(\mathbf{x}), \mathbf{y}; \boldsymbol{\theta})\right] \leq \left|\mathbb{E}_{\mathbf{x},\mathbf{y}}\left[l(\boldsymbol{\eta}(\mathbf{x}), \mathbf{y}; \boldsymbol{\theta})\right] - \frac{1}{|\mathcal{V}|}\sum_{i \in \mathcal{V}} l(\boldsymbol{\eta}(\mathbf{x}_i), \mathbf{y}_i; \boldsymbol{\theta}_{\mathcal{S}})\right| + \frac{1}{|\mathcal{V}|}\sum_{i \in \mathcal{V}} l(\boldsymbol{\eta}(\mathbf{x}_i), \mathbf{y}_i; \boldsymbol{\theta}_{\mathcal{S}})$$

The first term in the above equation is the difference between true generalization loss and the full set empirical loss which is inaccessible. Thus, we focus on the full set loss given model $\boldsymbol{\theta}_{\mathcal{S}}$. We assume that for every input $\mathbf{x}_i$ in the full set, there exists an $\mathbf{x}_j$ in the subset such that the training loss on $\mathbf{x}_j$ is 0 due to the optimization of the model on $\mathcal{S}$.

**Theorem 1** (Balanced Core-set Loss Bound). *Given the full set $\mathcal{V}$ and the subset $\mathcal{S}$, for each $\mathbf{x}_i \in \mathcal{V}$, we can locate a corresponding $\mathbf{x}_j \in \mathcal{S}$, such that $\|\mathbf{x}_j - \mathbf{x}_i\| = \min_{\mathbf{x}_n \in \mathcal{S}} \|\mathbf{x}_n - \mathbf{x}_i\|$ and $l(\boldsymbol{\eta}(\mathbf{x}_j), \mathbf{y}_j) = 0$. Then, we have*

$$\frac{1}{|\mathcal{V}|}\sum_{i \in \mathcal{V}} l(\mathbf{x}_i, \mathbf{y}_i, \boldsymbol{\theta}_{\mathcal{S}}) \leq \frac{1}{|\mathcal{V}|}\sum_{i \in \mathcal{V}} (\lambda^{\boldsymbol{\eta}}\|\mathbf{x}_i - \mathbf{x}_j\| + \lambda^y\|\mathbf{y}_i - \mathbf{y}_j\|) + L\sqrt{\frac{\log(1/\gamma)}{2|\mathcal{V}|}} \quad (1)$$

*with the probability of $1 - \gamma$, where $\lambda^{\boldsymbol{\eta}}$ and $\lambda^y$ are Lipschitz parameters, $L$ is the maximum possible loss and $\gamma$ is the probability of the Hoeffding's bound not holding true.*

*Proof Sketch.* To obtain the inequality, we utilize Hoeffding's bound. The problem then becomes finding the expectation of the full set loss ($\mathbb{E}[\frac{1}{|\mathcal{V}|}\sum_{i \in \mathcal{V}} l(\boldsymbol{\eta}(\mathbf{x}_i), \mathbf{y}_i; \boldsymbol{\theta}_{\mathcal{S}})]$). Note that unlike in the active learning case where the labels of the full set are unknown, we have access to both the inputs and labels in the subset selection scenario. Thus, we treat the model $\boldsymbol{\theta}_{\mathcal{S}}$ as the variable and convert all difference terms to $\|\mathbf{x}_i - \mathbf{x}_j\|$ or $\|\mathbf{y}_i - \mathbf{y}_j\|$ using Lipschitz conditions. The proof mainly involves the following step where we utilize the triangle inequality and the assumption that $l(\boldsymbol{\eta}(\mathbf{x}_j), \mathbf{y}_j; \boldsymbol{\theta}_{\mathcal{S}}) =$

$0, \forall \mathbf{x}_j \in \mathcal{S}$:

$$\mathbb{E}\left[\frac{1}{|\mathcal{V}|}\sum_{i \in \mathcal{V}} l(\boldsymbol{\eta}(\mathbf{x}_i), \mathbf{y}_i; \boldsymbol{\theta}_{\mathcal{S}})\right]$$

$$= \frac{1}{|\mathcal{V}|}\sum_{i \in \mathcal{V}} \mathbb{E}[l(\boldsymbol{\eta}(\mathbf{x}_i), \mathbf{y}_i; \boldsymbol{\theta}_{\mathcal{S}}) - l(\boldsymbol{\eta}(\mathbf{x}_j), \mathbf{y}_i; \boldsymbol{\theta}_{\mathcal{S}}) + l(\boldsymbol{\eta}(\mathbf{x}_j), \mathbf{y}_i; \boldsymbol{\theta}_{\mathcal{S}}) - l(\boldsymbol{\eta}(\mathbf{x}_j), \mathbf{y}_j; \boldsymbol{\theta}_{\mathcal{S}})]]$$

$$\leq \frac{1}{|\mathcal{V}|}\sum_{i \in \mathcal{V}} \mathbb{E}[l(\boldsymbol{\eta}(\mathbf{x}_i), \mathbf{y}_i; \boldsymbol{\theta}_{\mathcal{S}}) - l(\boldsymbol{\eta}(\mathbf{x}_j), \mathbf{y}_i; \boldsymbol{\theta}_{\mathcal{S}})| + |l(\boldsymbol{\eta}(\mathbf{x}_j), \mathbf{y}_i; \boldsymbol{\theta}_{\mathcal{S}}) - l(\boldsymbol{\eta}(\mathbf{x}_j), \mathbf{y}_j; \boldsymbol{\theta}_{\mathcal{S}})]] \quad (2)$$

$\square$

**Remark** (Decomposing Loss Bound: Feature and Label Objectives). *Eq. (1) gives the upper bound of the training loss, which has two main components: 1) the feature difference (or similarity) $\|\mathbf{x}_i - \mathbf{x}_j\|$ and 2) the label variability $\|\mathbf{y}_i - \mathbf{y}_j\|$. Thus, the optimal subset should be able to minimize both the feature difference and label variability to obtain a tight loss bound.*

**Bridging label variability with difficulty score.** Naturally, a diversity-guided approach will minimize the feature objective. Next, we will show that a difficulty-based approach can account for the label objective. In a typical difficulty-based selection, the score often aligns with the shape of the decision boundary. For example, the EL2N score is defined as $\mathbb{E}[\|\boldsymbol{\eta}(\mathbf{x}) - \mathbf{y}\|]$.

It highly correlates with the variance of prediction $\boldsymbol{\eta}(\mathbf{x})$ and whether the prediction is correct. Along the most difficult part of the joint data distribution, we can expect the data samples to have large variances of $\eta^{(k)}(\mathbf{x})$ produced by the model (especially a less overfitted model as the one used in our initial training) among all $K$ classes and are more likely to be misclassified. This means that the high difficulty scores will be distributed near the difficult part of the decision boundary. We can demonstrate this behavior with a synthetic dataset. The dataset is designed to have four moon-shaped classes, with slight overlapping (noises) as visualized in Figure 4 (d). As we show the EL2N score with a color map in Figure 2 (a), we can see

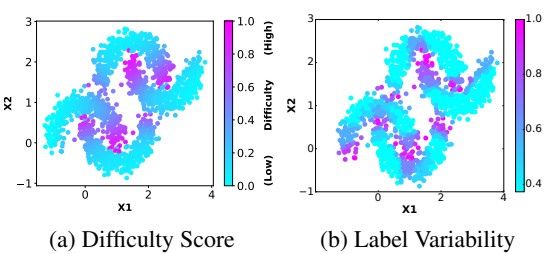

(a) Difficulty Score  (b) Label Variability

Figure 2: (a) The difficulty level of the samples for the synthetic dataset is computed using the EL2N score. The darker points refers to the difficult samples with higher value of EL2N score. The EL2N score is computed at epoch 10. (b) The expected label variability $\|\mathbf{y}_i - \mathbf{y}_j\|$ in the 10-sample neighborhood (scaled value).

exactly how the boundary points have the highest difficulty scores. Next, we will connect the EL2N score to the $\|\mathbf{y}_i - \mathbf{y}_j\|$ objective.

**Theorem 2** (EL2N lower bounds the label variability). *Assuming a subset sample $(\mathbf{x}_j, \mathbf{y}_j) \in \mathcal{S}$ is located in a difficult region (e.g., near the decision boundary), where (i) the neighborhood $\mathcal{N}_j$ is dense ($\|\mathbf{x}_j - \mathbf{x}_i\| \leq \delta_x, \forall(\mathbf{x}_i, \mathbf{y}_i) \in \mathcal{N}_j$ for $|\mathcal{N}_j|$ closest points) and (ii) the label variability is high ($p(\|\mathbf{y}_i - \mathbf{y}_j\| > 0) \geq \xi$), the EL2N score produced by a smooth model (such as the initial model $\boldsymbol{\eta}_0(x; \mathcal{V})$ that we use) will lower bound the label variability in this neighborhood $\mathcal{N}_j$.*

*Proof.* For the initial model trained for a few epochs on the full set $\mathcal{V}$, we denote it as $\boldsymbol{\eta}_0$. Given a difficult region as specified by the theorem, we consider the closest neighbors $\mathbf{x}_i$ and $\mathbf{x}_j$, which implies $\delta_x \approx 0$. Assume that $\mathbf{x}_j$ is correctly predicted: $\|\mathbf{y}_j - \boldsymbol{\eta}_0(\mathbf{x}_j)\| \approx 0$. Then, we have

$$\|\mathbf{y}_i - \mathbf{y}_j\| \approx \|\mathbf{y}_i - \mathbf{y}_j\| + \lambda^{\boldsymbol{\eta}_0}\delta_x \geq \|\mathbf{y}_i - \mathbf{y}_j - \boldsymbol{\eta}_0(\mathbf{x}_i) + \boldsymbol{\eta}_0(\mathbf{x}_j)\|$$
$$= \|(\mathbf{y}_i - \boldsymbol{\eta}_0(\mathbf{x}_i)) - (\mathbf{y}_j - \boldsymbol{\eta}_0(\mathbf{x}_j))\|$$
$$\approx \|\mathbf{y}_i - \boldsymbol{\eta}_0(\mathbf{x}_i)\| \quad (3)$$

If $\mathbf{x}_j$ is from a difficult region as specified by the theorem, then two conditions (i) and (ii) are satisfied. Consider another data sample $\mathbf{x}_j$ from the dense neighborhood where $\|\mathbf{x}_j - \mathbf{x}_i\| \leq \delta_x$,

we have $\|\boldsymbol{\eta}_0(\mathbf{x}_j) - \boldsymbol{\eta}_0(\mathbf{x}_i)\| \leq \lambda^{\boldsymbol{\eta}_0} \delta_x$. Since $\delta_x \to 0$, we have $\boldsymbol{\eta}_0(\mathbf{x}_j) \approx \boldsymbol{\eta}_0(\mathbf{x}_i)$. On the other hand, condition (ii) implies $\mathbf{y}_i \neq \mathbf{y}_j$, so $\mathbf{x}_i$ is likely to be wrongly predicted by the model. Then, the $\|\mathbf{y}_i - \mathbf{y}_j\|$ term can be lower bounded by the difference between the model prediction and label of the wrongly classified sample for the pair $(\mathbf{x}_i, \mathbf{x}_j)$: $\|\mathbf{y}_j - \boldsymbol{\eta}_0(\mathbf{x}_j)\|$. This way, in the most difficult region, we can approximate the overall $\|\mathbf{y}_i - \mathbf{y}_j\|$ by the expected difference between the prediction and the label. More importantly, if the subset is populated in the most difficult region, this lower bound will not change if we permute $(\mathbf{x}_j, \mathbf{y}_j)$ and $(\mathbf{x}_i, \mathbf{y}_i)$ in the same neighborhood $\mathcal{N}_j$ (as long as $p(\|\mathbf{y}_i - \mathbf{y}_j\| > 0) \geq \xi$) even if the wrongly classified samples are exchanged. We can then average the expected difference between the prediction and the label, which resembles the definition of the EL2N score $EL2N = \mathbb{E}_t[\|\boldsymbol{\eta}_0(x) - \mathbf{y}\|]$, where the expectation is taken over several undertrained initial models $\boldsymbol{\theta}_{\mathcal{V}}^{(t)}$. $\qquad\square$

In Figure 2 (b), we show the averaged label variability $|\sum_{i \in \mathcal{N}_j} \|\mathbf{y}_i - \mathbf{y}_j\|| / |\mathcal{N}_j|$ in a random 10-sample neighborhood setting (the values are scaled for visualization purpose). We can see that the trend of label variability matches the EL2N score in the difficult regions as shown in Figure 2 (a).

## 3.2 DIVERSITY-DIFFICULTY BALANCED ONE-SHOT SUBSET SELECTION

**Further analysis of the balanced core-set loss bound.** The balanced core-set loss bound given in (1) is comprised of two major components that correspond to feature similarity and label variability, respectively. For a subset with a fixed size, the bound reveals that a good subset should keep both terms small in order to obtain a tight loss bound. In this way, a model trained using such a subset can be expected to perform similarly as a model trained from the full set, which essentially optimize the l.h.s. of (1). Since the model property (especially $\lambda^{\boldsymbol{\eta}}$) is unknown, a principled and fine-grained mechanism is required to optimally balance these two components to make them jointly close to the full set loss. Further, as the subset size changes, the contribution from the two components may vary significantly, which in turn will affect the optimal balancing mechanism. In particular, when the subset size is small, the first term tends to dominate the entire bound because if some major clusters in the data distribution is completely missed, then all the data samples in the entire cluster will be represented by some dissimilar data samples from different clusters. This will accumulate a large feature difference that leads to a very loose bound. As the subset size increases and representative samples are properly chosen from all major clusters, the label variability starts to make a more obvious contribution to the overall bound. As revealed in our proof of Theorem 2, completely missing a difficult region will lead to a large label difference, which will result in a larger loss bound. Intuitively, missing samples from these regions will make the model lose the opportunity to learn a fine-grained decision boundary to further improve the generalization performance.

**Balancing diversity-difficulty through an expressive importance function.** In order to have fine-grained control for a balanced subset selection, we not only want to be able to control the target difficulty level but also how much the selection method will emphasize on the chosen level. In other words, we need a function to control both the *peak* and the *sharpness* of the selection. Accordingly, we design an *importance* function $\mathcal{I}(\cdot, \cdot)$ that includes two tunable parameters $c$ and $\alpha$, where $c$ controls the peak location w.r.t. the difficulty of data samples and $\alpha$ controls the sharpness:

$$\mathcal{I}(\mathbf{x}_j, \mathbf{y}_j) = \left[ \frac{\sin\left(\pi(D_j - c)\right) + 1}{2} \right]^{\alpha} \tag{4}$$

where $D_j \in [0, 1]$ denotes a difficulty metric such that $D_j \to 0$ ($D_j \to 1$) for easy (difficult) samples, $c \in [1.5, 2.5]$, and $\alpha \geq 0$ are the tunable parameters. A smaller value of $c$ gives importance to easier samples whereas a larger value of $c$ gives importance to the difficult samples. When $\alpha = 0$, the function is flat, assigning equal importance to all samples and increasing $\alpha$ will increase the sharpness. Figure 3 presents a set of importance functions with distinct shapes when fixing $\alpha = 1$ while varying $c$ or fixing $c = 2$ while varying $\alpha$. The blue

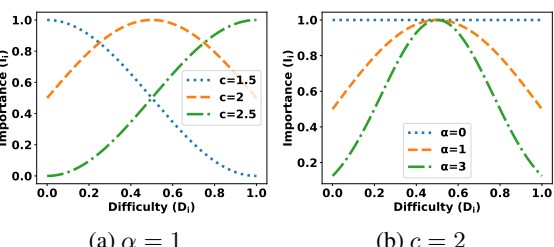

(a) $\alpha = 1$        (b) $c = 2$

Figure 3: Importance functions with fixed $\alpha = 1$ and $c = 2$

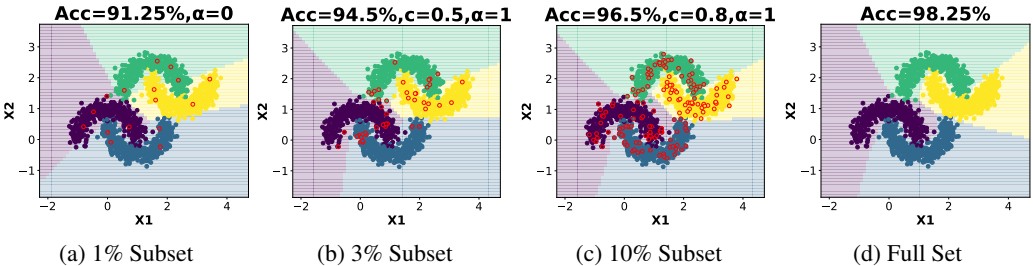

|(a) 1% Subset|(b) 3% Subset|(c) 10% Subset|(d) Full Set|

Figure 4: Subset selection for different subset sizes compared with the full set decision boundary.

(dot) plot in (a) gives priority to easier samples; the green (dash-dot) plot gives priority to difficult samples; and the orange (dash) plot gives priority to moderately difficult samples. In (b), the blue (dot) plot assigns uniform importance and the green (dash-dot) plot increases the sharpness of the curve by increasing the value of $\alpha$.

The rich expressiveness of the proposed importance function provides the flexibility to assign proper importance to difficult or easier samples depending on the required subset size, according to our analysis of the balanced loss bound. Hence, both the peak and sharpness parameters can be modeled as a function over the subset size. In general, the parameter $c$ increases as the subset size increases to give more focus to difficult samples. This is because we should focus on the more difficult samples and fine-tune the decision boundary when we can afford the additional budget. In contrast, for the small budget regime, we keep $\alpha$ small, making the function more flat and encouraging diversity-based selection. With a larger subset, $\alpha$ will not have a great impact as long as it is moderately large. We conduct experiments on a synthetic dataset to demonstrate how the proposed importance function can perform an optimally balanced subset selection as the size of the subset varies. The results are shown in Figure 4. Given the extremely small subset size (1%), it is preferred to let the model choose more diverse (and representative) data samples to cover a wide range of the data space by setting $\alpha = 0$. As the subset increases (3% → 10%), the peak of $\mathcal{I}$ can be shifted to higher difficulty levels by increasing both $\alpha$ and $c$. As can be seen, by training the model using a subset that is only 10% of the full set, it can discover a decision boundary as shown in (c) really close to the one using a model trained using the full set, as shown in (d).

**Balanced subset selection Function.** Combining the minimization of the diversity objective using the maximization of the similarity between the full set and the subset, and the minimization of the difficulty objective using our controllable importance function, we propose the balanced subset selection function as:

$$F(\mathcal{S}) = \sum_{i \in \mathcal{V}} \max_{j \in \mathcal{S}} \texttt{Sim}(\mathbf{x}_i, \mathbf{x}_j) \mathcal{I}(\mathbf{x}_j, \mathbf{y}_j) \tag{5}$$

where we use multiplication since we remain agnostic about the Lipschitz coefficients. Even with our fine-grained difficulty control, there is still the risk of selecting noises especially when we target the most difficult. To this end, we will adopt the cutoff mechanism as in (Zheng et al., 2023). In Eq. (5), the ranges of $c$ and $\alpha$ ensure the non-negativity of $\mathcal{I}$. Thus, $F(\mathcal{S})$ is a monotonically increasing function and can be shown to be submodular. This allows us to use a lazy greedy algorithm to approximate the optimum subset that can minimize $F(\mathcal{S})$. The greedy algorithm starts with an empty set $\mathcal{S} = \phi$ and keeps on adding samples $(\mathbf{x}_j, \mathbf{y}_j)$ to subset $\mathcal{S}$ that maximizes the gain:

$$F((\mathbf{x}_j, \mathbf{y}_j)|\mathcal{S}) = F(\mathcal{S} \cup (\mathbf{x}_j, \mathbf{y}_j)) - F(\mathcal{S}) \tag{6}$$

The pseudo code summarizing our implementation is described in Appendix C.

## 4 EXPERIMENTS

We conducted experiments on both synthetic and real-world data, aiming to further verify our proposed important theoretical results through the former and demonstrate the superior empirical performance through the latter. Limited by space, the synthetic experimental results are presented in Appendix D.1. Our real data experiments are conducted using four datasets: SVHN, CIFAR10, CIFAR100, and Tiny-ImageNet. We present both comparison result and a detailed ablation study.

**Comparison baselines.** We compare BOSS with seven baselines: **1) Random**: The samples are selected uniformly. **2) CRAIG**: CRAIG is one of the first representative-based subset selections

developed for classical models as well as deep learning models. It selects the subset by matching the gradient update signals of the full set and the subset (Mirzasoleiman et al., 2020). **3) GradMatch**: GradMatch uses orthogonal matching pursuit algorithm to match the gradient of subset and training set (Killamsetty et al., 2021a). **4) Adacore**: Adacore uses hessian preconditioned gradient instead of gradient (Pooladzandi et al., 2022). **5) LCMAT**: LCMAT selects the subset such that they match the loss curvature of the full set and the subset by matching the gradient and maximum eigenvalue of hessian between the full set and the subset (Shin et al., 2023). **6) Moderate**: Moderate core-set introduces distance-based scores such that samples with features closer to the median of the features of the related class is more important such that they keep the most important samples and prune the unimportant ones (Xia et al., 2023). **7) CCS**: CCS is coverage-centric core-set selection, which choose data samples randomly across different strata of importance scores giving priority to sparse strata (Zheng et al., 2023).

Table 1: Comparison results on subsets with different sizes

| Dataset | Subset | Random | CRAIG | GradMatch | Adacore | LCMAT | Moderate | CCS | BOSS(Ours) |
|---|---|---|---|---|---|---|---|---|---|
| Tiny ImageNet | 10% | $24.11_{\pm1.9}$ | $24.61_{\pm0.9}$ | $23.68_{\pm1.5}$ | $24.12_{\pm1.5}$ | $23.26_{\pm1.9}$ | $24.16_{\pm1.3}$ | $29.59_{\pm0.9}$ | $\mathbf{33.22}_{\pm0.5}$ |
| | 20% | $37.67_{\pm0.3}$ | $37.76_{\pm0.6}$ | $38.20_{\pm1.3}$ | $37.94_{\pm0.6}$ | $36.71_{\pm0.8}$ | $37.57_{\pm1.1}$ | $40.42_{\pm0.6}$ | $\mathbf{45.73}_{\pm0.4}$ |
| | 30% | $45.12_{\pm0.9}$ | $44.63_{\pm0.5}$ | $44.93_{\pm0.6}$ | $44.72_{\pm0.5}$ | $44.06_{\pm0.38}$ | $45.30_{\pm0.4}$ | $47.11_{\pm0.5}$ | $\mathbf{51.75}_{\pm0.4}$ |
| | 50% | $53.07_{\pm0.7}$ | $53.03_{\pm0.6}$ | $53.81_{\pm0.2}$ | $53.37_{\pm0.4}$ | $53.10_{\pm0.4}$ | $53.31_{\pm0.4}$ | $55.11_{\pm0.3}$ | $\mathbf{57.88}_{\pm0.2}$ |
| CIFAR 100 | 10% | $37.35_{\pm1.9}$ | $38.67_{\pm1.3}$ | $36.68_{\pm0.6}$ | $37.65_{\pm0.8}$ | $37.23_{\pm0.8}$ | $37.76_{\pm0.9}$ | $40.26_{\pm1.6}$ | $\mathbf{47.58}_{\pm0.5}$ |
| | 20% | $51.55_{\pm2.6}$ | $51.44_{\pm1.7}$ | $53.16_{\pm2.2}$ | $52.79_{\pm0.8}$ | $53.11_{\pm0.3}$ | $50.90_{\pm1.9}$ | $55.48_{\pm1.8}$ | $\mathbf{61.44}_{\pm0.7}$ |
| | 30% | $62.89_{\pm0.6}$ | $62.92_{\pm0.7}$ | $63.02_{\pm1.2}$ | $62.28_{\pm1.2}$ | $62.25_{\pm0.8}$ | $62.55_{\pm0.6}$ | $64.61_{\pm0.5}$ | $\mathbf{67.89}_{\pm0.2}$ |
| | 50% | $70.67_{\pm0.3}$ | $70.69_{\pm0.5}$ | $70.68_{\pm0.4}$ | $71.19_{\pm0.3}$ | $70.53_{\pm0.4}$ | $71.13_{\pm0.2}$ | $71.53_{\pm0.3}$ | $\mathbf{74.03}_{\pm0.3}$ |
| CIFAR 10 | 10% | $70.69_{\pm1.2}$ | $70.96_{\pm1.6}$ | $72.26_{\pm0.5}$ | $72.65_{\pm0.9}$ | $71.03_{\pm2.6}$ | $72.04_{\pm0.7}$ | $74.78_{\pm1.8}$ | $\mathbf{79.47}_{\pm0.5}$ |
| | 20% | $83.27_{\pm1.2}$ | $83.36_{\pm1.5}$ | $84.30_{\pm0.9}$ | $84.30_{\pm1.2}$ | $83.98_{\pm1.3}$ | $83.64_{\pm0.8}$ | $86.45_{\pm2.1}$ | $\mathbf{87.82}_{\pm0.9}$ |
| | 30% | $88.89_{\pm0.6}$ | $88.98_{\pm1.2}$ | $88.47_{\pm0.6}$ | $88.37_{\pm1.2}$ | $88.54_{\pm0.7}$ | $88.46_{\pm0.5}$ | $91.49_{\pm0.5}$ | $\mathbf{92.15}_{\pm0.6}$ |
| | 50% | $92.69_{\pm0.2}$ | $92.75_{\pm0.3}$ | $91.89_{\pm0.4}$ | $92.67_{\pm0.5}$ | $92.58_{\pm0.2}$ | $92.61_{\pm0.2}$ | $93.45_{\pm0.5}$ | $\mathbf{94.36}_{\pm0.2}$ |
| SVHN | 8% | $84.98_{\pm1.9}$ | $84.30_{\pm1.1}$ | $84.31_{\pm1.8}$ | $82.31_{\pm2.6}$ | $84.05_{\pm1.8}$ | $84.51_{\pm0.7}$ | $86.69_{\pm1.5}$ | $\mathbf{89.52}_{\pm0.8}$ |
| | 12% | $87.16_{\pm2.4}$ | $88.49_{\pm0.4}$ | $88.99_{\pm1.0}$ | $88.41_{\pm1.3}$ | $87.49_{\pm1.3}$ | $88.97_{\pm0.6}$ | $92.16_{\pm0.9}$ | $\mathbf{93.18}_{\pm0.5}$ |
| | 16% | $90.47_{\pm0.7}$ | $89.92_{\pm0.9}$ | $90.42_{\pm0.8}$ | $90.34_{\pm0.8}$ | $90.16_{\pm0.6}$ | $90.35_{\pm1.1}$ | $93.87_{\pm0.5}$ | $\mathbf{94.31}_{\pm0.3}$ |
| | 20% | $91.64_{\pm0.7}$ | $92.13_{\pm0.3}$ | $91.56_{\pm0.4}$ | $91.95_{\pm0.8}$ | $91.36_{\pm0.4}$ | $91.30_{\pm0.9}$ | $94.38_{\pm0.5}$ | $\mathbf{95.08}_{\pm0.3}$ |

**Experimental setup.** Our experiment setup follows existing approaches, such as (Shin et al., 2023; Guo et al., 2022; Zheng et al., 2023), where to select the subset, we first initialize a model by training it using the full dataset for a limited number of epochs. Then using the training dynamics obtained from the initialized model, we obtain the difficulty score for each sample which is used to select the subset. We then evaluate the selected subset by keeping the subset fixed and using the subset to train a new randomly initialized model. For the difficulty score, we mainly experiment using the EL2N score because it can be computed efficiently early on during the training. The features, gradients, and Hessians are computed from the second-last layer of the network. The baselines vary in the way they select the subset. For the model, we train the ResNet18 model (He et al., 2016) using SGD with a learning rate decay of $5 \times 10^{-4}$, starting learning rate of 0.1, and momentum of 0.9. We use ResNet34 for the Tiny ImageNet dataset. We use a batch size of 256. To compute the EL2N score, we use the training dynamics up to the first 10 epochs of the initial training. Similarly, we use the feature representations, gradients, and Hessians of epoch 10 of the initial training. The reported results are averaged over five runs. For our method, we sample the subset in a class-balanced fashion. Additional details of experimental setup can be found in the Appendix.

**Performance comparison.** Table 1 show the result for the four datasets as compared to the baselines. Our method systematically integrates both diversity and difficulty while performing a balanced selection according to the subset size and the nature of dataset. As a result, it significantly outperforms all the competitive baselines, especially on the low budget regime. The performance difference decreases as the subset size increases because there is less room for improvement.

**Impact of key parameters.** In order to show the behavior of the importance function, we run experiments over different values of $c$ and $\alpha$ and present the results in Figure 5 (a) and (b). Their optimal values depend on the subset size and the complexity of the dataset. The parameters $c$, and $\alpha$ control the importance given to certain difficulty levels in their own unique ways. For smaller subset sizes, the best value of $c$ is lower whereas it is higher for a larger subset. This is equivalent to giving more importance to difficult samples when the subset size is larger and more importance to easier samples when the subset size is smaller. For $\alpha$, when we have a larger subset size, by increasing the value of $\alpha$, we can increase the sharpness of the importance function to select more points from the difficult region. Additional results are presented in the Appendix.

Table 2: Ablation study results

| Dataset | Subset | Diversity | Difficulty (EL2N) | Diversity + Difficulty | Diversity + Difficulty + Cutoff |
|---|---|---|---|---|---|
| Tiny ImageNet | 10% | $24.04_{\pm 1.1}$ | $3.39_{\pm 0.3}$ | $32.12_{\pm 0.9}$ | $\mathbf{33.22}_{\pm 0.5}$ |
| | 20% | $37.63_{\pm 0.9}$ | $7.75_{\pm 0.5}$ | $43.49_{\pm 0.6}$ | $\mathbf{45.73}_{\pm 0.4}$ |
| | 30% | $44.47_{\pm 0.9}$ | $20.92_{\pm 1.9}$ | $48.39_{\pm 0.5}$ | $\mathbf{51.75}_{\pm 0.4}$ |
| | 50% | $52.78_{\pm 0.2}$ | $44.42_{\pm 0.6}$ | $54.83_{\pm 0.1}$ | $\mathbf{57.88}_{\pm 0.2}$ |
| CIFAR 100 | 10% | $36.69_{\pm 0.6}$ | $7.11_{\pm 0.4}$ | $\mathbf{47.68}_{\pm 0.8}$ | $47.58_{\pm 0.5}$ |
| | 20% | $52.04_{\pm 0.9}$ | $14.78_{\pm 0.5}$ | $59.66_{\pm 0.8}$ | $\mathbf{61.44}_{\pm 0.7}$ |
| | 30% | $62.41_{\pm 0.3}$ | $31.99_{\pm 1.1}$ | $66.60_{\pm 0.7}$ | $\mathbf{67.89}_{\pm 0.2}$ |
| | 50% | $70.18_{\pm 0.1}$ | $65.73_{\pm 1.0}$ | $72.35_{\pm 0.3}$ | $\mathbf{74.03}_{\pm 0.3}$ |
| CIFAR 10 | 10% | $72.17_{\pm 0.9}$ | $22.26_{\pm 0.4}$ | $78.45_{\pm 0.5}$ | $\mathbf{79.47}_{\pm 0.5}$ |
| | 20% | $84.10_{\pm 1.0}$ | $41.95_{\pm 1.9}$ | $85.82_{\pm 0.4}$ | $\mathbf{87.82}_{\pm 0.9}$ |
| | 30% | $88.63_{\pm 0.4}$ | $78.75_{\pm 6.4}$ | $88.81_{\pm 0.4}$ | $\mathbf{92.15}_{\pm 0.6}$ |
| | 50% | $92.52_{\pm 0.5}$ | $94.41_{\pm 0.2}$ | $\mathbf{94.42}_{\pm 0.2}$ | $94.36_{\pm 0.2}$ |
| SVHN | 8% | $83.96_{\pm 2.5}$ | $63.00_{\pm 1.9}$ | $87.99_{\pm 0.5}$ | $\mathbf{89.52}_{\pm 0.8}$ |
| | 12% | $89.02_{\pm 1.2}$ | $77.68_{\pm 1.2}$ | $91.97_{\pm 0.7}$ | $\mathbf{93.18}_{\pm 0.5}$ |
| | 16% | $89.83_{\pm 0.9}$ | $81.83_{\pm 1.7}$ | $93.45_{\pm 0.3}$ | $\mathbf{94.31}_{\pm 0.3}$ |
| | 20% | $91.27_{\pm 0.6}$ | $84.84_{\pm 1.1}$ | $93.39_{\pm 0.4}$ | $\mathbf{95.08}_{\pm 0.3}$ |

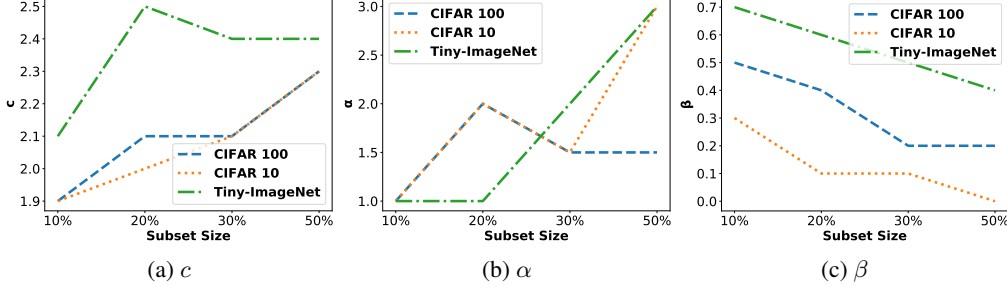

(a) $c$  (b) $\alpha$  (c) $\beta$

Figure 5: The impact of key parameters with respect to the subset size

To ensure a fair comparison with CCS, we also leverage the cutoff rate parameter $\beta$. Figure 5(c) shows that $\beta$ should be set higher for a small subset size to avoid choosing noisy or outlier samples. This can ensure a more robust subset of data samples to be selected.

**Ablation study.** Our ablation study investigates the following parts: 1) the Diversity component, where we minimize the distance between $\mathbf{x}_i$ and $\mathbf{x}_j$; 2) the Difficulty component, where we select samples based on their difficulty scores; 3) Diversity+Difficulty, which performs sample selection based on the proposed balanced subset selection function $F$; and 4) Diversity+Difficulty+Cutoff, where we further prune the potential noisy examples while balancing diversity and difficulty. Table 2 shows the ablation study results. Only using the Diversity component, which selects the representative samples has sub-optimal performance since it does not consider any difficulty-level information of the datasets. Furthermore, only using the Difficulty component has the worst performance, especially for the low budget regime. This is because the selection is highly biased towards those difficult samples, which causes a large feature difference, leading to a very loose loss bound, as our analysis shows. When combining the difficulty and diversity through the proposed importance function, the performance improves by a large margin. Integrating the cutoff mechanism can slightly improve the performance, especially for those more complex datasets, such as Tiny ImageNet. This is because those datasets may likely contain noisy or outlier samples, which if selected, could negatively impact the quality of the subset.

## 5 CONCLUSION

Subset selection is a promising direction in solving the problem of increasing resource consumption by large deep learning models. Existing subset selection methods have limitations because their selection criteria do not consider a joint distribution of data diversity and difficulty. We propose a novel subset selection strategy that systematically integrates both diversity and difficulty supported by a balanced core-set loss bound. The novel loss bound also suggests important relationship between the difficulty of the selected sample and the subset size, which leads to an expressive importance function that enables us to select appropriate samples according to the subset size. Our theoretical analysis along with the empirical results on synthetic and real-world data demonstrate the greater effectiveness of BOSS compared with the competitive baselines.

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

# Appendix

Appendix A summarizes the major notations used in the main paper. Appendix B provides detailed proofs of our main theoretical results. Appendix C describes our subset selection algorithm. Appendix D provides additional experimental details, link to the source code, results for synthetic data, and ablation studies on real-world data. Appendix E summarizes the limitations and societal impact of the proposed method.

## A    SUMMARY OF NOTATIONS

Table 3: Summary of Notations

| Symbol | Description |
|:---:|:---|
| $\mathcal{V}$ | Set of all the training samples (Full Set) |
| $\mathcal{S}$ | Set of samples that are selected (Subset) |
| $\lambda$ | Lipschitz parameter |
| $\boldsymbol{\eta}$ | Neural network regression function |
| $\mathbf{x}_i$ | Input feature of a sample in the Full-set |
| $\mathbf{y}_i$ | True label of a sample in the Full-set |
| $\mathbf{x}_j$ | Input feature of a sample in the Subset |
| $\mathbf{y}_j$ | True label of a sample in the Subset |
| $\boldsymbol{\theta}_{\mathcal{S}}$ | Model trained on the Subset |
| $l(\cdot)$ | Loss function |
| $\gamma$ | Probability for Hoeffding's inequality |
| $L$ | Upper bound for the loss function |
| $C$ | Number of classes |
| $F(\cdot)$ | Balanced subset selection function |
| $Sim$ | Similarity function |
| $\mathcal{I}(\mathbf{x}_j, \mathbf{y}_j)$ | Importance score of sample $j$ |
| $D_j$ | Difficulty score of sample $j$ |
| $c$ | Parameter controlling peak of importance function |
| $\alpha$ | Parameter controlling the sharpness of importance function |
| $\beta$ | Hard cut-off rate |

## B    PROOFS OF THEORETICAL RESULTS

In this section, we provide detailed proofs for the proposed theorems in the main paper. We have introduced the balanced core-set loss bound in Section 3.1. Here we first expand the loss bound derivation and analysis and then provide detailed proofs for Theorem 1.

### B.1    OUR TAKE ON THE CORE-SET LOSS BOUND

In (Sener & Savarese, 2018), the authors proposed the classic core-set cover loss bound, in the active learning setting. The first step is to upper bound the true expectation of generalization loss by the full set loss, which is shown in:

$$\mathbb{E}_{x,y}\left[l(\boldsymbol{\eta}(\mathbf{x}), \mathbf{y}; \boldsymbol{\theta})\right] \leq \left| \mathbb{E}_{x,y}\left[l(\boldsymbol{\eta}(\mathbf{x}), \mathbf{y}; \boldsymbol{\theta})\right] - \frac{1}{|\mathcal{V}|}\sum_{i \in \mathcal{V}} l(\boldsymbol{\eta}(\mathbf{x}_i), \mathbf{y}_i; \boldsymbol{\theta}_{\mathcal{S}}) \right|$$

$$+ \frac{1}{|\mathcal{V}|}\sum_{i \in \mathcal{V}} l(\boldsymbol{\eta}(\mathbf{x}_i), \mathbf{y}_i; \boldsymbol{\theta}_{\mathcal{S}}) \tag{7}$$

Same as (Sener & Savarese, 2018) and (Zheng et al., 2023), we adopt this approach and focus on the expectation of the full set loss. However, differently from their approach, we formally tailor the problem in the known full set setting. In our case, all data samples $(\mathbf{x}_i, \mathbf{y}_i) \in \mathcal{V}$ are treated as known, and the unknown is the model $\boldsymbol{\theta}_{\mathcal{S}}$ trained on the subset (and the corresponding outputs

$\boldsymbol{\eta}(\mathbf{x}_i)$), which corresponds accurately to the subset selection problem. For the same reason, we denote the loss function by $l(\boldsymbol{\eta}(\mathbf{x}_i), \mathbf{y}_i; \boldsymbol{\theta}_\mathcal{S})$. Next, we also apply the Hoeffding's bound to analyze the full set loss. However, unlike (Sener & Savarese, 2018) and (Zheng et al., 2023) which reached loose conclusions by bounding the feature difference $\|\mathbf{x}_i - \mathbf{x}_j\|$ using the core-set cover, we consider the joint effect over both the feature and the label, arriving at a balanced core-set loss bound which will be explained below.

### B.2 Proof for Theorem 1

*Proof.* We obtain the inequality in Eq. (1) by applying the Hoeffding's bound:

If $X_1, X_2, ..., X_n$ are independent and $a_i \le X_i \le b_i$ almost surely, then the sum $S_n = X_1 + ... + X_n$ satisfy

$$P(S_n - \mathbb{E}[S_n] \ge t) \le \exp\left(-\frac{2t^2}{\sum_{i=1}^n (b_i - a_i)^2}\right) \tag{8}$$

In Theorem 1, we apply the above inequality to the full set loss $\sum_{i\in\mathcal{V}} l(\boldsymbol{\eta}(\mathbf{x}_i), \mathbf{y}_i; \boldsymbol{\theta}_\mathcal{S})$ by substituting $S_n = \sum_{i\in\mathcal{V}} l(\boldsymbol{\eta}(\mathbf{x}_i), \mathbf{y}_i; \boldsymbol{\theta}_\mathcal{S})$ and using $0 \le l(\boldsymbol{\eta}(\mathbf{x}_i), \mathbf{y}_i; \boldsymbol{\theta}_\mathcal{S}) \le L$ ($L$ being the maximum loss value):

$$P\left(\frac{1}{|\mathcal{V}|}\sum_{i\in\mathcal{V}} l(\boldsymbol{\eta}(\mathbf{x}_i), \mathbf{y}_i; \boldsymbol{\theta}_\mathcal{S}) - \mathbb{E}\left[\frac{1}{|\mathcal{V}|}\sum_{i\in\mathcal{V}} l(\boldsymbol{\eta}(\mathbf{x}_i), \mathbf{y}_i; \boldsymbol{\theta}_\mathcal{S})\right] \ge \frac{t}{|\mathcal{V}|}\right) \le \exp\left(-\frac{2t^2}{|\mathcal{V}|L^2}\right) \tag{9}$$

Let $\gamma = \exp\left(-\frac{2t^2}{|\mathcal{V}|L^2}\right)$, then with probability $1 - \gamma$,

$$\frac{1}{|\mathcal{V}|}\sum_{i\in\mathcal{V}} l(\boldsymbol{\eta}(\mathbf{x}_i), \mathbf{y}_i; \boldsymbol{\theta}_\mathcal{S}) - \mathbb{E}\left[\frac{1}{|\mathcal{V}|}\sum_{i\in\mathcal{V}} l(\boldsymbol{\eta}(\mathbf{x}_i), \mathbf{y}_i; \boldsymbol{\theta}_\mathcal{S})\right] \le \frac{t}{|\mathcal{V}|} \tag{10}$$

Rearranging and solving for $t$, we get

$$\frac{1}{|\mathcal{V}|}\sum_{i\in\mathcal{V}} l(\boldsymbol{\eta}(\mathbf{x}_i), \mathbf{y}_i; \boldsymbol{\theta}_\mathcal{S}) - \mathbb{E}\left[\frac{1}{|\mathcal{V}|}\sum_{i\in\mathcal{V}} l(\boldsymbol{\eta}(\mathbf{x}_i), \mathbf{y}_i; \boldsymbol{\theta}_\mathcal{S})\right] \le L\sqrt{\frac{\log(1/\gamma)}{2|\mathcal{V}|}} \tag{11}$$

Next, we explain the expectation of the full set loss and obtain the balanced combination result:

$$\begin{aligned}
&\mathbb{E}\left[\frac{1}{|\mathcal{V}|}\sum_{i\in\mathcal{V}} l(\boldsymbol{\eta}(\mathbf{x}_i), \mathbf{y}_i; \boldsymbol{\theta}_\mathcal{S})\right] \\
&= \frac{1}{|\mathcal{V}|}\sum_{i\in\mathcal{V}} \mathbb{E}[|l(\boldsymbol{\eta}(\mathbf{x}_i), \mathbf{y}_i; \boldsymbol{\theta}_\mathcal{S}) - l(\boldsymbol{\eta}(\mathbf{x}_j), \mathbf{y}_i; \boldsymbol{\theta}_\mathcal{S}) + l(\boldsymbol{\eta}(\mathbf{x}_j), \mathbf{y}_i; \boldsymbol{\theta}_\mathcal{S}) - l(\boldsymbol{\eta}(\mathbf{x}_j), \mathbf{y}_j; \boldsymbol{\theta}_\mathcal{S})|] \\
&\le \frac{1}{|\mathcal{V}|}\sum_{i\in\mathcal{V}} \mathbb{E}[|l(\boldsymbol{\eta}(\mathbf{x}_i), \mathbf{y}_i; \boldsymbol{\theta}_\mathcal{S}) - l(\boldsymbol{\eta}(\mathbf{x}_j), \mathbf{y}_i; \boldsymbol{\theta}_\mathcal{S})| + |l(\boldsymbol{\eta}(\mathbf{x}_j), \mathbf{y}_i; \boldsymbol{\theta}_\mathcal{S}) - l(\boldsymbol{\eta}(\mathbf{x}_j), \mathbf{y}_j; \boldsymbol{\theta}_\mathcal{S})|] \\
&= \frac{1}{|\mathcal{V}|}\sum_{i\in\mathcal{V}} \bigg( \mathbb{E}[|l(\boldsymbol{\eta}(\mathbf{x}_i), \mathbf{y}_i; \boldsymbol{\theta}_\mathcal{S}) - l(\boldsymbol{\eta}(\mathbf{x}_j), \mathbf{y}_i; \boldsymbol{\theta}_\mathcal{S})|] \\
&\quad + \mathbb{E}[|l(\boldsymbol{\eta}(\mathbf{x}_j), \mathbf{y}_i; \boldsymbol{\theta}_\mathcal{S}) - l(\boldsymbol{\eta}(\mathbf{x}_j), \mathbf{y}_j; \boldsymbol{\theta}_\mathcal{S})|]\bigg)
\end{aligned} \tag{12}$$

which has been broken into two terms.

For the first term $\mathbb{E}[|l(\boldsymbol{\eta}(\mathbf{x}_i), \mathbf{y}_i; \boldsymbol{\theta}_\mathcal{S}) - l(\boldsymbol{\eta}(\mathbf{x}_j), \mathbf{y}_i; \boldsymbol{\theta}_\mathcal{S})|]$, we utilize the Lipschitzness of the model combined with the Lipschitzness of the loss function to get $\mathbb{E}[|l(\boldsymbol{\eta}(\mathbf{x}_i), \mathbf{y}_i; \boldsymbol{\theta}_\mathcal{S}) - l(\boldsymbol{\eta}(\mathbf{x}_j), \mathbf{y}_i; \boldsymbol{\theta}_\mathcal{S})|] \le \mathbb{E}[\lambda^{\boldsymbol{\eta}}\|\mathbf{x}_i - \mathbf{x}_j\|]$. The expectation should be taken over the model prediction $\eta^{(k)}(\mathbf{x}_i)$, and will result in a class-irrelevant term for $\sum_k \eta^{(k)}(\mathbf{x}_i) = 1$ if we use loss functions like the cross-entropy loss so that $\lambda^{\eta^{(k)}}$ is the same for all $k$.

For the second term $\mathbb{E}[|l(\boldsymbol{\eta}(\mathbf{x}_j), \mathbf{y}_i; \boldsymbol{\theta}_\mathcal{S}) - l(\boldsymbol{\eta}(\mathbf{x}_j), \mathbf{y}_j; \boldsymbol{\theta}_\mathcal{S})|]$ we directly use the Lipschitzness of the loss function w.r.t. $\mathbf{y}$ and it is independent from $\boldsymbol{\eta}(\mathbf{x}_i)$ so we have $\lambda^y\|\mathbf{y}_i - \mathbf{y}_j\|$.

Substituting all the above terms back and we will obtain Eq. (1). $\qquad\square$

### B.3 ANALYSIS OF THEOREM 2

In Theorem 2, we connect the EL2N score to the expected label variability in a neighborhood $\mathcal{N}_j$ of a subset point $(\mathbf{x}_j, \mathbf{y}_j)$. Unlike Theorem 1, which is more general about the loss on full set $\mathcal{V}$, this connection is specifically made in the difficult region.

**Remark** (2.1). *Why is it important to consider the difficult region?*

We have defined the difficult region as dense and having high label variability. This is because we really focus on the disadvantage of only considering the diversity aspect or the difficulty aspect. Intuitively, if the true distribution of $(\mathbf{x}, \mathbf{y})$ is clearly separated and smooth, using a few samples can perfectly explain the classification problem as long as all classes are represented. However, if there exists a difficult region such that $\|\mathbf{x}_j - \mathbf{x}_i\| \leq \delta_x$ and $p(\|\mathbf{y}_i - \mathbf{y}_j\| > 0) \geq \xi$, then it poses a challenge for both single-sided approaches: a diversity-only approach can not identify informative data samples that help learn the decision boundary in this difficult region, while a difficulty-only approach will be highly biased towards this region and won't be able to efficiently represent the majority of samples which are easy to classify. We will present more visualizations using the synthetic dataset in Appendix D.1. Thus, it takes a delicate balancing to improve the overall objective in (1) when the difficult region exists.

**Remark** (2.2). *How do we utilize the EL2N score to explain the label variability?*

In the proof of Theorem 2, we present an approximately lower bounding relationship between the label difference between $\mathbf{y}_i$ and $\mathbf{y}_j$ and the EL2N score of the wrongly classified sample $(\mathbf{x}_j, \mathbf{y}_j)$. If we assume that the neighborhood $\mathcal{N}_j$ includes $(\mathbf{x}_i, \mathbf{y}_i)$, $(\mathbf{x}_j, \mathbf{y}_j)$, and $\{(\mathbf{x}_n, \mathbf{y}_n)\}_{n=1}^{|\mathcal{N}_j|-2}$. With $p \geq \xi$, we have $\mathbf{y}_i \neq \mathbf{y}_j$, thus flipping them will likely flip the model prediction for all $\mathbf{x}_n$. When we take the expectation over all samples in $\mathcal{N}_j$, the averaged $\|\mathbf{y}_i - \mathbf{y}_j\|$ will be connected to the distribution of the EL2N scores of these data samples which is the expectation over a series of models $\boldsymbol{\theta}_\mathcal{V}^{(t)}$.

## C ALGORITHM

---

**Algorithm 1** BOSS (Balanced One-Shot Subset Selection)

Initial Training

**Input**: Dataset $\mathcal{V}$
**Output**: Difficulty score $D_i$, feature vector $\mathbf{x}_i$

1: Initialize full set model $\boldsymbol{\theta}_\mathcal{V}$
2: Train $\boldsymbol{\theta}_\mathcal{V}$ on $\mathcal{V}$
3: From $\boldsymbol{\theta}_\mathcal{V}$ obtain $\boldsymbol{\eta}(\mathbf{x}_i^t)$, $\mathbf{y}_i^t$ for each epoch $t \in [1, T_0]$
4: From $\boldsymbol{\theta}_\mathcal{V}$ obtain $\mathbf{x}_i$ for epoch $T_0$.
5: Compute EL2N Score $D_i$ using $\boldsymbol{\eta}(\mathbf{x}_i^t)$, $\mathbf{y}_i^t$
6: **return** $D_i, \mathbf{x}_i$

Subset Selection

**Input**: Dataset $\mathcal{V}$, Subset size $b$, difficulty score $D_i$, input feature $x_i$
**Parameter**: Hard example prune rate $\beta$, importance function parameters $c$ and $\alpha$
**Output**: Subset $\mathcal{S}$

1: $I_i \leftarrow ((\sin(\pi(D_i - c)) + 1)/2)^\alpha$ {Convert difficulty score to importance score}
2: $\mathcal{V}' \leftarrow \mathcal{V} \setminus \{|\mathcal{V}| * \beta \text{ hardest samples}\}$ {Prune hardest samples}
3: $\mathcal{S} \leftarrow \phi$ {Start with an empty subset and add to the subset until we reach the budget for the subset:}
4: **while** $|\mathcal{S}| < b * |\mathcal{V}|$ **do**
5:     $F(\mathcal{S}) \leftarrow \sum_{i \in \mathcal{V}} \max_{j \in \mathcal{S}} \texttt{Sim}(\mathbf{x}_i, \mathbf{x}_j) I_j$ {Compute the facility location function from the feature similarity and sample importance}
6:     $j \in \text{argmax}_{e \in \mathcal{V} \setminus \mathcal{S}} F(e|\mathcal{S})$ {Using lazy-greedy algorithm, select sample $j$ which gives us the maximum conditional gain $F(e|\mathcal{S})$}
7:     $\mathcal{S} \leftarrow \mathcal{S} \cup \{j\}$ {Update the subset with the new element}
8: **end while**
9: **return** $\mathcal{S}$

---

Our method has three main components, 1) Initial training where we first train a model to generate training dynamics from which we can compute the importance scores, 2) Generating importance score from the training dynamics, and 3) Selecting Subset and evaluating the subset by training a new model using the selected subset. Details are provided in Algorithm 1.

# D  ADDITIONAL EXPERIMENTAL DETAILS AND RESULTS

We perform our experiments on machines with GPUs: A100, V100, and P4. In our experiments, when combining the difficulty and diversity, we show the results for using the EL2N score (Paul et al., 2021) as a difficulty metric. The EL2N score is calculated in the initial training phase by averaging the error norm over the first 10 epochs. To leverage the cutoff, the Accumulated Margin (AUM) metric (Pleiss et al., 2020) is leveraged, for which we need to train the model for the full epoch (200 epoch for CIFAR10 and CIFAR100, and 100 epoch for Tiny ImageNet and SVHN). Our implementation details and source code can be found here.

## D.1  SYNTHETIC DATA EXPERIMENTS

We create the synthetic data to include four moons with two input features such that we can visualize and simulate the complex decision boundary. The synthetic data is visualized in the Figure 3. The dataset has 2000 samples which are split into 80/20 train/test sets. For the model, we use a fully connected neural network with two hidden layers each containing 100 neurons. To train the neural network we use Adam optimizer with a learning rate of 0.001, $\epsilon$ = 1e-08, and weight decay = 0. For the full set, we train the model for 100 epochs. The model reaches a test accuracy of 98.25% while training on the full data.

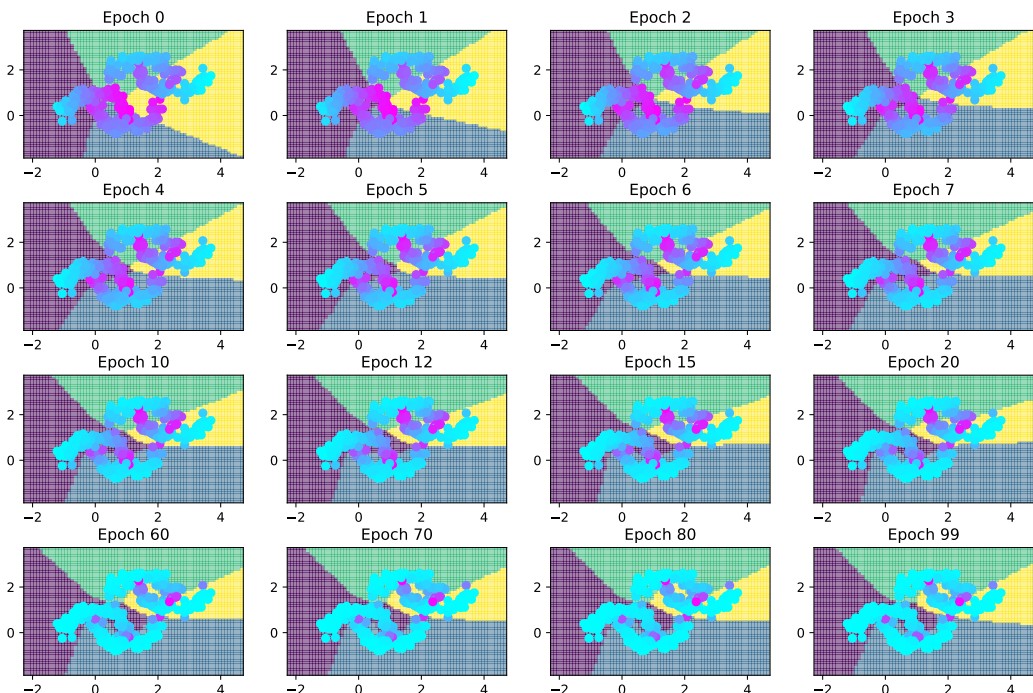

Figure 6: Decision Boundary Evolution

**Decision boundary evolution.** Figure 6 shows the evolution of the decision boundary along with the difficulty level of each data point for every epoch. The difficulty is calculated using the EL2N score which is the L2 norm of prediction and the onehot label and averaged over the previous epochs. As we train the model for a higher number of epochs, the model is able the learn complex decision boundaries or the curved region and most of the samples become easier or has low EL2N score. However, the difficulty score computed at the earlier epochs, for instance at epoch 10, truly captures

the difficulty level of samples along the decision boundary. This agrees with the past methods which compute the EL2N score at epoch 10.

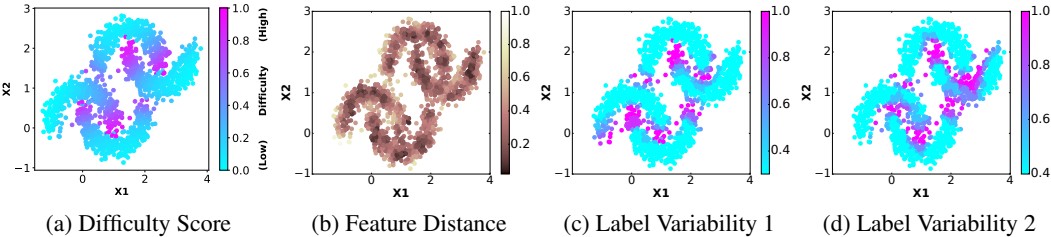

(a) Difficulty Score      (b) Feature Distance      (c) Label Variability 1      (d) Label Variability 2

Figure 7: Difficulty score, feature distance, and label variability comparison. All values are scaled to 0 to 1 for better visualization. (c) Label variability 1 and (d) Label variability 2 are permutations of Figure 2(b) by randomly changing the data samples included in the 10-sample neighborhood.

**Label variability visualization** Following Appendix B.3, we visualize the terms that have been discussed in our theoretical results using the synthetic dataset.

In Figure 7, we visualize the difficulty score, feature distance, and label variability with random permutations to the anchor points being used as $(\mathbf{x}_j, \mathbf{y}_j)$ in the 10-sample neighborhood case. From Figure 7 (a) and (b), we see that the difficulty score does not correlate with the feature distance, and the feature distance is not informative in the difficult region, where the distance is consistently low because it is the denser area. From Figure 2 (b) and Figure 7 (c) and (d), we can see that even with different permutations, the label variability shares the same trend as the difficulty score near the decision boundary.

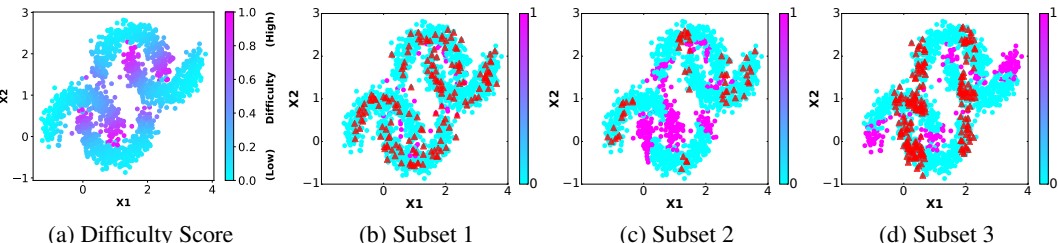

(a) Difficulty Score      (b) Subset 1      (c) Subset 2      (d) Subset 3

Figure 8: Difficulty score and $\mathbb{1}(\mathbf{y}_i \neq \mathbf{y}_j; j = \arg\max_{n \in \mathcal{S}} \texttt{Sim}(\mathbf{x}_n, \mathbf{x}_i))$ comparison, where red triangles represent data samples in $\mathcal{S}$.

In Figure 8, we show a different visualization presenting the actual label differences between the full set and the subset if we choose from different regions. Figure 8 (b) shows a diverse selection, while Figure 8 (c) and (d) show two different biased selections. In all cases, the data samples near the decision boundary have a different label from the nearest sample in $\mathcal{S}$ ($\mathbb{1}(\mathbf{y}_i \neq \mathbf{y}_j; j = \arg\max_{n \in \mathcal{S}} \texttt{Sim}(\mathbf{x}_n, \mathbf{x}_i))$=1). This further supports our motivation as allocating the budget to cover the more difficult region does not guarantee the reduction of the label variability objective in the loss bound given in (1) unless we can cover all these samples. Thus, it is important that we propose the balanced selection function.

**Impact of the subset size to the balanced core-set bound.** In the main paper, we state that the subset size will affect the optimal diversity-difficulty balance: in data data-scarce regime, the diversity dominates while as the subset budget increases, more difficult samples should be picked. To more clearly show how the subset size impacts the balanced core-set loss bound, we quantify and visualize the two major components in the loss bound: $\sum_i \|\mathbf{x}_i - \mathbf{x}_j\|$ and $\sum_i \|\mathbf{y}_i - \mathbf{y}_j\|$, which essentially captures the feature distance and label distance between the selected subset and the full set, respectively. As can be seen from Figure 9 (a), for a small subset size, when choosing the subset based on the label variability (or difficulty), it can help to quickly reduce the label distance. However, it also leads to a very large feature distance that makes the overall bound large. Figure 9 (b) further confirms this because the selected samples misses some major regions of the data distribution as stated in the main paper. In contrast, when focusing on choosing samples based on the

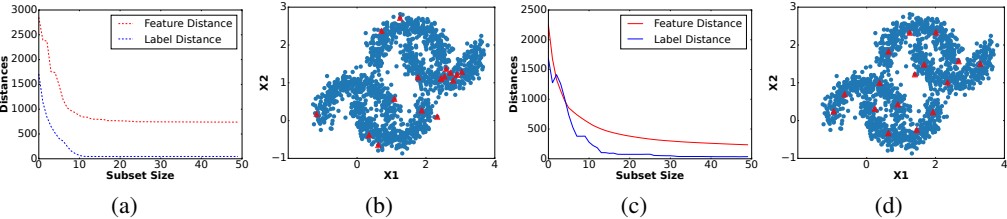

Figure 9: Trend of the two major components in the balanced core-set loss bound: (a) feature and label distance trends by label variability based selection; (b) first 16 samples selected based on label variability; (c) feature and label distance trends by focusing on diversity first; (d) first 16 samples selected based on feature similarity

first component (*i.e.,* diversity), the feature distance drops significantly as shown in Figure 9 (c), which implies that the selected subset can represent the entire data distribution well. This is further confirmed by Figure 9 (d), which visualizes the distribution of the selected data samples. As more samples are selected, they will start to cover the difficult regions, which can effectively bring down the label distance as shown in Figure 9 (c).

**BOSS Subset selection comparison.** Here we include the subset selection visualization and comparison for the synthetic dataset. The red circles are the samples selected in the subset. We already saw the comparison of CSS and BOSS for the 10% subset in Figure 1. Here we further compare these methods for 1% and 3% subset sizes. In all the cases, BOSS outperforms CCS.

Figure 10 compares the subset selected by CCS and BOSS for 1% subset size. In this case, BOSS can select the diverse samples by setting $\alpha = 0$. Although the model cannot learn the complex decision boundary because of the lack of enough data, the CSS misses samples from the important regions and learns an even worse decision boundary.

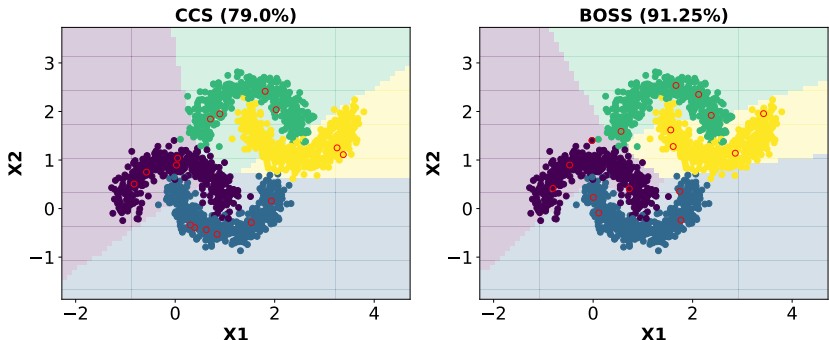

Figure 10: CCS vs BOSS for 1% Subset Size for the synthetic data. For CCS, $\beta = 0.1$. For BOSS, $\beta = 0$ and $\alpha = 0$ which is the same as only using representative-based selection.

Similarly, Figure 11 compares CCS and BOSS for a 3% subset size. This figure also verifies that the CCS misses the samples from the critical region that our method is able to capture. In turn our method learns the complex decision boundary to achieve better performance than the CCS baseline.

Figure 12 shows the comparison of the subset selected by the representative-based method which matches the feature of the subset and the full set (Diverse) compared with the subset selected by our method. The representative-based subset selection does not consider the sample difficulty which leads it to ignore samples from a very difficult region. However, our method is able to give more emphasis on the difficult region to better learn the decision boundary.

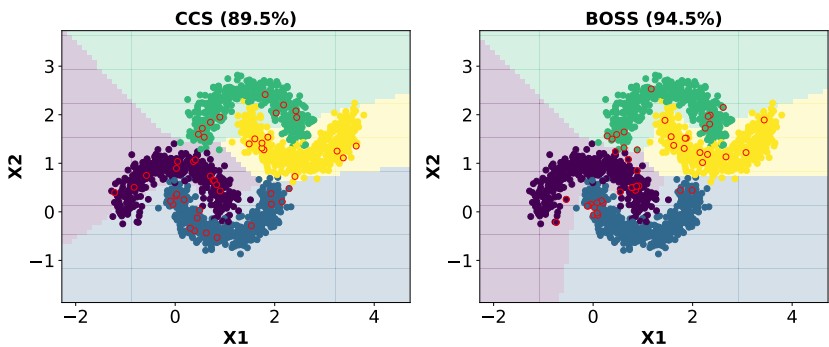

Figure 11: CCS vs BOSS for 3% Subset Size for the synthetic data. For CCS, $\beta = 0.1$. For BOSS, $\beta = 0$, $c = 0.5$, and $\alpha = 1$

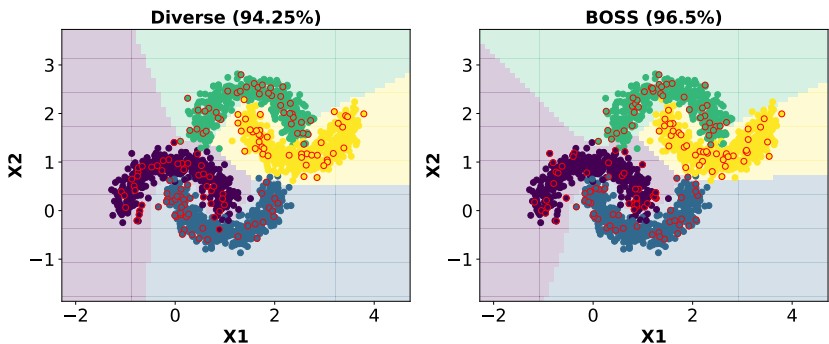

Figure 12: Visualization of representative-based subset selection compared with our method. The red circles are the points selected for the subset. The subset size is 10% and $\beta = 0$. For Diverse, $\alpha = 0$ and for BOSS, $\alpha = 1$, and $c = 0.8$.

## D.2 ADDITIONAL ABLATION STUDY

Table 4: Comparison of CCS and BOSS for Tiny ImageNet while using different difficulty metrics.

| Subset | EL2N | | Forgetting | | AUM | |
|---|---|---|---|---|---|---|
| | CCS | BOSS | CCS | BOSS | CCS | BOSS |
| 10% | $29.59_{\pm 0.9}$ | $\mathbf{33.22}_{\pm 0.5}$ | $30.44_{\pm 1.7}$ | $\mathbf{33.78}_{\pm 1.3}$ | $31.51_{\pm 1.2}$ | $\mathbf{33.47}_{\pm 0.7}$ |
| 20% | $40.42_{\pm 0.6}$ | $\mathbf{45.73}_{\pm 0.4}$ | $42.75_{\pm 0.9}$ | $\mathbf{45.56}_{\pm 0.6}$ | $42.05_{\pm 0.4}$ | $\mathbf{45.80}_{\pm 0.6}$ |
| 30% | $47.11_{\pm 0.5}$ | $\mathbf{51.75}_{\pm 0.4}$ | $48.61_{\pm 0.7}$ | $\mathbf{51.81}_{\pm 0.2}$ | $48.92_{\pm 0.1}$ | $\mathbf{52.11}_{\pm 0.3}$ |
| 50% | $55.11_{\pm 0.3}$ | $\mathbf{57.88}_{\pm 0.2}$ | $55.91_{\pm 0.5}$ | $\mathbf{57.82}_{\pm 0.3}$ | $55.74_{\pm 0.4}$ | $\mathbf{57.79}_{\pm 0.3}$ |

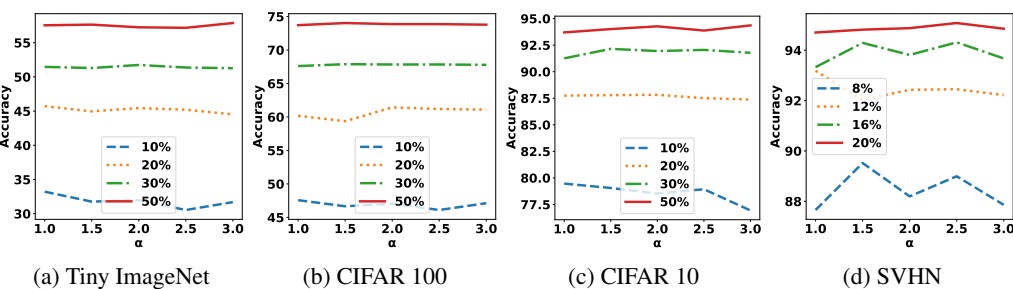

(a) Tiny ImageNet      (b) CIFAR 100      (c) CIFAR 10      (d) SVHN

Figure 13: The impact on performance when $c$ is fixed while varying the value of $\alpha$.

**Other difficulty metrics.** Table 4 shows the comparison between CCS and BOSS while using EL2N (Paul et al., 2021), Forgetting (Toneva et al., 2019), and Accumulated Margin(AUM) (Pleiss et al., 2020) for the difficulty metric. For a fair comparison, we compare CCS and BOSS for each difficulty metric separately. Our method (BOSS) outperforms CCS for every difficulty metric.

**Impact of $\alpha$.** In the main paper, we have already discussed the impact of the key parameters. Although we have clear guidance for setting $c$ as it is closely related to the subset size $|\mathcal{S}|$, the selection of best $\alpha$ is less explicit. However, we can see that in general we still prefer smaller $\alpha$ in the small subset size case because we want the importance function to be flat and better focus on diversity-based selection, and we prefer moderately larger $\alpha$ to allow more expressive selection based on the difficulty. In Figure 13, we show the change in performance as we change the parameter $\alpha$ and keep the parameter $c$ fixed to the best value for the given subset size and the dataset. For the larger subset sizes, the change in $\alpha$ has a minor effect on the performance when compared to smaller subset sizes, and less obviously for the more difficult datasets (Tiny ImageNet and CIFAR 100) where the difficulty levels may have a good separation. Thus, we can verify that $\alpha$ is mostly useful for the fine-tuning of subset selection especially given a small subset size, and becomes less sensitive for larger subset sizes.

Table 5: EfficientNet-B0 results for CIFAR100

| Subset | Random | Moderate | CCS | BOSS |
|--------|--------|----------|-----|------|
| 10% | $30.51_{\pm1.0}$ | $32.59_{\pm1.3}$ | $36.91_{\pm2.2}$ | $\mathbf{42.64}_{\pm0.6}$ |
| 20% | $43.52_{\pm1.9}$ | $42.04_{\pm2.2}$ | $46.53_{\pm3.7}$ | $\mathbf{53.39}_{\pm0.3}$ |
| 30% | $55.48_{\pm0.7}$ | $55.26_{\pm1.7}$ | $56.89_{\pm0.3}$ | $\mathbf{60.37}_{\pm0.4}$ |
| 50% | $64.05_{\pm0.7}$ | $63.91_{\pm0.3}$ | $63.59_{\pm0.5}$ | $\mathbf{68.27}_{\pm0.5}$ |

Table 6: ViT-B16 results for CIFAR100

| Subset | Random | Moderate | CCS | BOSS |
|--------|--------|----------|-----|------|
| 10% | $78.49_{\pm0.7}$ | $50.41_{\pm0.7}$ | $78.62_{\pm0.3}$ | $\mathbf{79.97}_{\pm0.4}$ |
| 20% | $81.87_{\pm0.7}$ | $69.81_{\pm0.5}$ | $81.95_{\pm0.8}$ | $\mathbf{83.19}_{\pm0.1}$ |
| 30% | $83.98_{\pm0.2}$ | $77.66_{\pm0.5}$ | $84.93_{\pm0.1}$ | $\mathbf{85.08}_{\pm0.1}$ |
| 50% | $85.88_{\pm0.1}$ | $84.19_{\pm0.0}$ | $85.88_{\pm0.1}$ | $\mathbf{86.55}_{\pm0.1}$ |

### D.2.1 RESULTS FOR OTHER MODELS

In Tables 5 and 6, we evaluate our method on two additional models: EfficientNet-B0 (Tan & Le, 2019) and a vision transformer ViT-B16 (Dosovitskiy et al., 2021). For EfficientNet, we use our previously mentioned setting. For ViT, we use pre-trained weights, batch size of 128, learning rate of 0.01, and train for 12 epochs. We compare with the two most recent and competitive baselines (CCS (Zheng et al., 2023)), Moderate (Xia et al., 2023)) and Random. Our method performs better than the baselines for both models. The performance margin for ViT is lower because we are using pre-trained weights and the room for improvement is small. Nonetheless, we show the usefulness of our method for other models than ResNet.

Table 7: Imbalanced CIFAR100 using exponential decay

| Subset | Random | Moderate | CCS | BOSS |
|--------|--------|----------|-----|------|
| 10% | $27.39_{\pm0.9}$ | $25.37_{\pm1.7}$ | $29.41_{\pm0.5}$ | $\mathbf{36.63}_{\pm0.9}$ |
| 20% | $42.82_{\pm1.1}$ | $40.57_{\pm0.6}$ | $44.36_{\pm1.7}$ | $\mathbf{50.91}_{\pm0.6}$ |
| 30% | $52.51_{\pm0.7}$ | $50.00_{\pm3.0}$ | $50.87_{\pm1.4}$ | $\mathbf{57.15}_{\pm0.5}$ |
| 50% | $63.03_{\pm0.6}$ | $61.76_{\pm0.2}$ | $61.86_{\pm0.5}$ | $\mathbf{66.72}_{\pm0.1}$ |

### D.2.2 IMBALANCED DATA

Tables 7 and 8 show the result for class imbalanced data. We make CIFAR100 imbalanced in two ways. First (7), by using exponential decay $N_{c_i} \times e^{-0.01i}$ where $N_{c_i}$ is the number of samples

Table 8: Imbalanced CIFAR100 using step

| Subset | Random | Moderate | CCS | BOSS |
|--------|--------|----------|-----|------|
| 10% | $31.66_{\pm 0.7}$ | $27.02_{\pm 0.7}$ | $33.60_{\pm 0.9}$ | $\mathbf{38.72}_{\pm 0.9}$ |
| 20% | $47.36_{\pm 0.9}$ | $41.77_{\pm 2.9}$ | $46.82_{\pm 0.6}$ | $\mathbf{53.75}_{\pm 0.3}$ |
| 30% | $56.64_{\pm 0.2}$ | $52.31_{\pm 0.6}$ | $52.81_{\pm 1.1}$ | $\mathbf{58.37}_{\pm 0.4}$ |
| 50% | $62.19_{\pm 0.9}$ | $59.96_{\pm 0.7}$ | $60.25_{\pm 0.2}$ | $\mathbf{66.03}_{\pm 0.2}$ |

for class $c_i$. Second (8), by pruning 80% of data from 20% of classes. Our method has better performance compared to the baselines in both of the settings. It is interesting to note that random is also competitive and even better in some cases compared to the other baselines.

Table 9: Time comparison in seconds.

| Dataset | Subset Size | Subset Selection (Initial Training) | Subset Selection (Selection Algorithm) | Subset Training | Full Set Training |
|---------|-------------|-------------------------------------|----------------------------------------|-----------------|-------------------|
| CIFAR100 | 10% | 219 | 9 | 346 | 4387 |
|          | 20% |     | 13 | 587 |      |
|          | 30% |     | 14 | 800 |      |
|          | 50% |     | 15 | 1816 |     |
| CIFAR10  | 10% | 173 | 11 | 342 | 3468 |
|          | 20% |     | 19 | 571 |      |
|          | 30% |     | 22 | 801 |      |
|          | 50% |     | 27 | 1244 |     |

### D.2.3 TIME COMPARISON

In Table 9, we compare the time taken by our method for *Subset Selection* and *Subset Training*. The *Subset Selection* consists of initial training for 10 epochs on the full data, and the lazy greedy algorithm to select the subset. The *Subset Selection* time is shorter compared to training on the subset (*Subset Training*) and also takes a very short time compared to training on the full set (*Full Set Training*). The time for the subset selection algorithm (time excluding the initial training) is significantly small compared to the initial training time and also does not require GPU computation. We measure the time in seconds using NVIDIA RTX A6000 GPU for CIFAR10 and CIFAR100 datasets.

## E    LIMITATIONS AND SOCIETAL IMPACT

In this paper, we have proposed a balanced one-shot subset selection method (BOSS). While using the subset reduces the resource consumption by large deep learning models, we should consider the change of information from the full set to the subset and the potential biases the selection might introduce. We should also carefully deploy the balanced selection method as data diversity and informativeness (difficulty) are being explicitly controlled. In real-world applications, only preserving the subset could be beneficial if the above aspects are well-considered, and can prove to be useful in scenarios such as continual learning and various socially impactful deep learning applications.

