# OpenReview forum: "BOSS: Diversity-Difficulty Balanced One-Shot Subset Selection for Data-Efficient Deep Learning"
_ICLR.cc/2024/Conference — Submitted to ICLR 2024_

### Official Review · Reviewer_AoWq · 2023-10-31

**Soundness:** 3 good
**Presentation:** 3 good
**Contribution:** 3 good
**Rating:** 5
**Confidence:** 4

**Summary:**

This paper inspects the coreset selection problem. A balanced core-set loss bound is first established to depict the generalization loss of the model trained on the subset. The authors claim that the bound is composed of two terms, one corresponds to the “diversity/coverage” of the coreset, and the other counts for the “difficulty” of the samples. The bound naturally unifies the diversity-based as well as the difficulty-based works developed previously, and the paper further provides an expressive importance function to optimally balance them. The authors find that the optimal balance is related to the subset size. In the data-scarce regime, the subset is supposed to be representative enough (diverse), while in the data-abundance regime, difficult samples are preferred. The resulting coreset selection strategy is named diversity-difficulty Balanced One-shot Subset Selection (BOSS), Experiments on both synthetic and real datasets are conducted to justify the effectiveness of the proposed method.

**Strengths:**

1.	Utilizing coreset selection to improve data efficiency is important for machine learning practices. The paper may be valuable to the community trying to address this problem.

2.	The paper is clearly written, and the authors do a good job presenting their intuitions developing the method.

3.	I appreciate the efforts the authors made connecting the core-set loss bound, subset diversity, and sample difficulty, which naturally unified the diversity-based as well as the difficulty-based works developed in previous literature.

4.	Experiments are conducted on both synthetic dataset and real-world datasets, validating the effectiveness of the proposed method in certain settings.

**Weaknesses:**

1.	Rather than rigorously derived from the balanced core-set loss bound, equation (5) seems to be simply a hand-crafted heuristic selection strategy combining the diversity-based method and the difficulty-based method. In theroem2, the authors claim that EL2N lower bound the label variability in difficult regions. I wonder if this holds for other regions as EL2N/difficulty is universally used in Equation (5). Besides, to minimize Equation (1), for the label variability term, we should minimize something upper bounds $|| \boldsymbol{y}_i -  \boldsymbol{y}_j ||$ instead of something lower bounds it like EL2N.

2.	The authors claim that the subset size will affect the optimal diversity-difficulty balance, in data data-scarce regime, the diversity dominates while as the subset budget increases, more difficult samples should be picked. While intuitively true and the authors give intuitive explanations, I can’t directly justify the statement directly from the core-set loss bound. More discussion will greatly strengthen the paper.

I will be happy to increase my score if the problems are addressed.

**Questions:**

Please see the weakness part above

---

> ### Author Response · Authors · 2023-11-18
>
> Thank you for the constructive feedback.
>
> **Q1: Rather than rigorously derived from the balanced core-set loss bound, Equation (5) seems to be simply a hand-crafted heuristic selection strategy combining the diversity-based method and the difficulty-based method**
>
> We would like to clarify that Equation 5 closely follows our theoretical result in Theorem 1 that decomposes the overall loss bound into the input similarity and label variability. As mentioned in our general response, the involvement of the label variability makes our method substantially different from existing methods (e.g., those based on feature/gradient similarity or the standard facility location function as commented by other reviewers). Another major novelty also comes from the importance function $I(\cdot)$ that goes beyond a difficulty based metric. It has a much richer functional form that connects the label variability (difficulty) and subset size. This allows us to achieve a dynamic balance between the input similarity and label variability based upon the subset size.  The importance function also bridges the theoretical objective of loss bound given in Theorem 1 and the empirical objective of submodular maximization of Equation 5. It is worth to note that this connection is essential because Equation 1 cannot be directly implemented as the model property ($\lambda^{\boldsymbol{\eta}}$) is unknown.
>
> **In theorem 2, the authors claim that EL2N lower bound the label variability in difficult regions. I wonder if this holds for other regions as EL2N/difficulty is universally used in Equation (5)**
>
> The *label variability* is mostly important when we want to select samples from difficult regions or points near the decision boundary. When we move towards the easier region both the EL2N score and label variability are significantly lower compared to the difficult region (as verified in the visualization of synthetic datasets).  So among the easier samples, we only focus on selecting representative samples, and the values of label variability will be small anyway.
>
> **Q2: Besides, to minimize Equation (1), for the label variability term, we should minimize something upper bounds $||{\bf y}_i-{\bf y}_j||$ instead of something lower bounds it like EL2N**
>
> This is a great question! Please refer to our general response regarding "Connecting label variability with EL2N score".
>
> **Q3: The authors claim that the subset size will affect the optimal diversity-difficulty balance, in data data-scarce regime, the diversity dominates while as the subset budget increases, more difficult samples should be picked. While intuitively true and the authors give intuitive explanations, I can’t directly justify the statement directly from the core-set loss bound**
>
> Thank you for this insightful question! To more clearly show how the subset size impacts the balanced core-set loss bound, we conduct additional experiments on the synthetic data, aiming to quantify and visualize the two major components in the loss bound: $\sum_i||{\bf x}_i-{\bf x}_j||$ and $\sum_i||{\bf y}_i-{\bf y}_j||$, which essentially captures the feature distance and label distance between the selected subset and the full set, respectively. As can be seen from Figure 9 (a) in the Appendix of the revised paper, for a small subset size, when choosing the subset based on the label variability (or difficulty), it can help to quickly reduce the label distance. However,  it also leads to a very large feature distance that makes the overall bound large. Figure 9 (b) further confirms this because the selected samples misses some major regions of the data distribution as stated in the main paper. In contrast, when focusing on choosing samples based on the first component (i.e., diversity), the feature distance drops significantly as shown in Figure 9 (c), which implies that the selected subset can represent the entire data distribution well. This is further confirmed by  Figure 9 (d), which visualizes the distribution of the selected data samples. As more samples are selected, they will start to cover the difficult regions, which can effectively bring down the label distance as shown in Figure 9 (c).

---

> ### Comment · Reviewer_AoWq · 2023-11-21
> **Thanks for the clarification**
>
> Thanks for the clarification, which has addressed most of my questions.  I like the endeavor made to combine the diversity-based method and the difficulty-based method. I understand the intuition relating the bound to the final method, However, I still feel that the method is not directly (rigorously) derived from the theoretical analysis.

---

> > ### Author Response · Authors · 2023-11-21
> >
> > We sincerely appreciate your confirmation that we have effectively addressed the majority of your inquiries. Additionally, we extend our gratitude for supporting our idea of combining diversity-based and difficulty-based methods.
> >
> > Regarding the last point, mathematically, the feature term monotonically decreases as the subset size increases, whereas the label term and the relative scale of the label term to the feature term are unknown without rigorous assumptions about the data-label distribution.  We tactfully address the issue by using the importance function to achieve the diversity-difficulty balance. It is a great challenge to directly model the feature term and label term, and looking at each of them separately does not help (as we have shown with the added content in the rebuttal).
> >
> > We would like to think that our method is clearly derived from the theoretical analysis. It is also worth noting that the upper bound is a common theoretical practice in most cases but can not be actually calculated. We’d like to know the reviewer’s opinion of what a directly (rigorously) derived method looks like (an example would be helpful).

---

### Official Review · Reviewer_BNoZ · 2023-10-31

**Soundness:** 2 fair
**Presentation:** 3 good
**Contribution:** 2 fair
**Rating:** 5
**Confidence:** 3

**Summary:**

One major drawback of standard subset selection is that the subset cannot accurately reflect the join data distribution. To tackle this drawback, BOSS aim to construct an optimal subset for data-efficient traning.
Samples are chosen for the subset with the goal of minimizing a balanced core-set loss bound.
A trade-off exists between feature similarity and label variability in the balanced core-set loss bound. To this end, it can take into account subset size, data type, variety, and difficulty.

**Strengths:**

- The proposed method is supported by prior evidence and is well stated.
- They balance the variety and difficulty of subset selection given a subset size.
- There are considerable performance improvements using the proposed methods

**Weaknesses:**

- For a fixed number of epochs, the entire dataset must be used to train a model.
- Absence of variety in experiments. ResNet is insufficient on its own to verify the efficacy of the proposed method. To validate their approach, it is necessary to conduct experiments on more models.
- There is no comparison between the entire train duration and the time required to generate a subset. The problem with the proposed process is that all of the data must be trained so that authors should perform experiments with computation complexity.

**Questions:**

- It is difficult to discern what the author intended when they write, "missed some critical regions(upper middle area)", as Figure 1(a) on page 2.
- What does the symbol gamma represent in Theorem.1 on page 4?
- What is the rationale behind the paper's assertion that "CCS still does not strike the right balance between diversity and the difficulty of subset selection"?

---

> ### Author Response · Authors · 2023-11-18
>
> Thank you for the constructive feedback.
>
> **Q1: the entire dataset must be used to train a model for a fixed number of epochs**
>
> We would like to clarify that it is a common practice for one-shot subset selection methods to perform initial training on full data for the first few epochs. This is followed by all the major one-shot subset methods, including CCS, LCMAT, Moderate, EL2N, and Forgetting. Such initial full-set training is essential to obtain necessary information (e.g., gradients, features, EL2N scores) about the data before subset selection can be performed. Furthermore, since the initial training only takes a few epochs, it introduces limited overhead (see our response to Q3 for details).
>
> When compared with the dynamic subset selection, although a small number of samples is used to train the model at a time, the dynamic subset selection will be conducted using a large fraction of data throughout the training. This is not feasible in applications such as continual learning where the full data is accessible only once.
>
> **Q2: experiments on more models**
>
> Thank you for this great suggestion! We have conducted additional experiments on both EfficientNet [1] and ViT [2]. Overall, we have used four models ResNet18 (for SVHN, CIFAR10, and CIFAR100), ResNet34 (for TinyImageNet), EfficientNet, and ViT (for CIAFR100). We are able to show the effectiveness and consistent trend of our method as compared with competitive baselines for all of these models.
>
> [1] Tan, Mingxing, and Quoc Le. "Efficientnet: Rethinking model scaling for convolutional neural networks." Proceedings of the 36th International Conference on Machine Learning, PMLR 97:6105-6114, 2019.
>
> [2] Dosovitskiy, Alexey, et al. "An Image is Worth 16x16 Words: Transformers for Image Recognition at Scale." International Conference on Learning Representations. 2021.
>
> ViT Result (CIFAR100):
> | Subset | Random | Moderate | CCS | BOSS |
> |--------|--------|----------|-----|------|
> | 10%    | 78.49 ± 0.7| 50.41 ± 0.7| 78.62 ± 0.3| **79.97 ± 0.4**|
> | 20%    | 81.87 ± 0.7| 69.81 ± 0.5| 81.95 ± 0.8| **83.19 ± 0.1**|
> | 30%    | 83.98 ± 0.2| 77.66 ± 0.5| 84.93 ± 0.1| **85.08 ± 0.1**|
> | 50%    | 85.88 ± 0.1| 84.19 ± 0.0| 85.88 ± 0.1| **86.55 ± 0.1**|
>
> Efficient Net Result (CIFAR100):
> | Subset | Random | Moderate | CCS | BOSS |
> |--------|--------|----------|-----|------|
> | 10%    | 30.51 ± 1.0| 32.59 ± 1.3| 36.91 ± 2.2| **42.64 ± 0.6**|
> | 20%    | 43.52 ± 1.9| 42.04 ± 2.2| 46.53 ± 3.7| **53.39 ± 0.3**|
> | 30%    | 55.48 ± 0.7| 55.26 ± 1.7| 56.89 ± 0.3| **60.37 ± 0.4**|
> | 50%    | 64.05 ± 0.7| 63.91 ± 0.3| 63.59 ± 0.5| **68.27 ± 0.5**|
>
> **Q3: comparison between train time and selection time; experiments with computation complexity**
>
> We compare the time taken for *Subset Selection*, and *Subset Training*. The *Subset Selection* consists of initial training for 10 epochs on the full set, and the lazy greedy algorithm to select the subset. The *Subset Selection* time is shorter compared to training on the subset (*Subset Training*) and is much more efficient compared to training on the full set (*Full Set Training*). The time for the subset selection algorithm (time excluding the initial training) is significantly small compared to the initial training time and this step does not require GPU computation.
> We measure the time in seconds using NVIDIA RTX A6000 GPU for CIFAR10 and CIFAR100 datasets.
>
> Time comparison results
> | Dataset   | Subset Size | Subset Selection (Initial Training) | Subset Selection (Selection Algorithm) | Subset Training | Full Set Training |
> |-----------|-------------|-------------------------------------|-----------------------------------------|------------------|-------------------|
> | CIFAR100  | 10%         |                                     | 9                                   | 346              |                   |
> |           | 20%         | 219                                 | 13                                  | 587              | 4387              |
> |           | 30%         |                                     | 14                                  | 800              |                   |
> |           | 50%         |                                     | 15                                  | 1816             |                   |
> | CIFAR10   | 10%         |                                     | 11                                  | 342              |                   |
> |           | 20%         | 173                                 | 19                                  | 571              | 3468              |
> |           | 30%         |                                     | 22                                  | 801              |                   |
> |           | 50%         |                                     | 27                                  | 1244             |                |
>
> **To be continued...**

---

> ### Author Response · Authors · 2023-11-18
>
> **Continuing...**
>
> **Q4: meaning of "missed some critical regions", Figure 1(a)**
>
> The critical regions refer to the regions near the boundary. We have annotated those regions in Figure 1 on page 2 of the revised paper.
>
> **Q5: what does $\gamma$ represent in Theorem 1?**
>
> This term is related to Hoeffding's bound. $\gamma$ is the probability that the bound does not hold true.  $1 - \gamma$ is the probability that the bound holds true.
>
> **Q6: rationale behind "CCS still does not strike the right balance between diversity and difficulty"**
>
> As we have shown with the theoretical analysis (please see our general response for more details), the optimal selection requires targeting different difficulty levels according to the subset size, which is impossible for CCS.

---

> ### Author Response · Authors · 2023-11-21
>
> We appreciate your feedback, and we hope our responses have addressed your queries and clarified any confusion related to our work. Following your suggestion, we conducted experiments on additional models beyond ResNet. Additionally, in an attempt to show the computational complexity, we have included a table comparing the time taken to select and train on the subset. The main paper has been revised in accordance with your recommendations. We hope that these answers meet your expectations, and we kindly ask for your consideration in updating your assessment. Please let us know if you have any further concerns, we would be more than happy to provide any additional information.

---

### Official Review · Reviewer_jBvt · 2023-11-06

**Soundness:** 3 good
**Presentation:** 3 good
**Contribution:** 3 good
**Rating:** 5
**Confidence:** 5

**Summary:**

The proposed method tackles the problem of data efficient subset selection. They claim that existing methods underperform in terms of generalization since they aim to find subsets that are either diverse or difficult. They propose a new technique called BOSS (diversity-difficulty Balanced One-shot Subset Selection) which aims to find an optimal subset that faithfully represent the joint data distribution which is comprised of both feature and label information. They do so by optimizing a novel balanced core-set loss.

**Strengths:**

- The paper is well written and clearly illustrates the underlying problem and the proposed solution.
- The paper covers a good chunk of related work in Sec 1
- The experiments are on multiple datasets
- Ablations studies help answer trade offs between diversity, difficulty and cutoff.

**Weaknesses:**

My main concern is the novelty of the work which can be improved by reinforcing the effectiveness of the proposed method. A few questions and suggestions are as follows:

- The proposed function is very similar to the standard facility location function, which is $\sum_{i \in V} max_{j \in A} Sim(x_i, x_j).$ The function additionally has the I(.) term which is the main contribution in my opinion. To fully understand the effect of the additional I(.) term, the authors should compare with the facility location submodular function.

- The authors discuss multiple relevant papers in this work but do not add comparison with many of them in the experiments. It would be great to compare with a few more method, e.g., Grad Match.

- The 'balanced' aspect of the proposed loss is still not clear to me. It would be imperative to add some experiments to show how the selected subsets are balanced. It would be even better if the authors can show some experiments on class imbalanced data. Most datasets currently in the experiments barely have any imbalance, which makes this analysis difficult.

**Questions:**

- Questions are mainly listed in the weaknesses section. Please refer them.

---

> ### Author Response · Authors · 2023-11-18
>
> Thank you for the constructive feedback.
>
> **Q1: The novelty of the work**
>
> Please refer to the general response as we distinguish the proposed method from the standard facility location function.
>
> **Q2: To fully understand the effect of the additional $I(\cdot)$ term, the authors should compare with the facility location submodular function.**
>
> Thank you for the suggestion. In the ablation study, we present subset selection using only the diversity component, which is the same as the standard submodular facility location function as suggested by the reviewer. Our result shows that by maximizing the combination of the *standard facility location function* (diversity) and the *importance function* (function of difficulty and subset size), we can achieve significant improvement compared to only using the *standard facility location submodular function* (diversity).
>
> Additionally, we want to emphasize that the importance function $I(\cdot)$ itself is not the only main contribution of our work. Our contribution also lies in the theoretical work bringing together the input similarity, label variability, and subset size, which justifies the need to maximize the joint objective of the importance function and submodular function (Equation 5) in order to minimize the loss bound in Equation 1.
>
> We also compare with other baselines such as CRAIG and Adacore which are also based on the submodular location function but use the gradients instead of the features.
>
> **Q3: Compare with more method, e.g., Grad Match**
>
> Following the reviewer's suggestion, we have compared with two additional baselines, including GradMatch and Adacore. The complete table is updated in the revised paper (see Table 1 on page 8). Our proposed method shows a clear advantage over these two baselines on all settings, which further justifies its effectiveness. We would also like to clarify that the representative-based subset selection methods are not initially used for one-shot subset selection but rather for dynamic subset selection. Therefore, the performance cannot be directly compared to the respective papers as we are evaluating them in a one-shot setting.
>
> | Dataset       | Subset | GradMatch | Adacore | BOSS(Ours) |
> |---------------|--------|-----------|---------|------------|
> | Tiny ImageNet | 10%    | 23.68 ± 1.5| 24.12 ± 1.5| **33.22 ± 0.5**|
> |               | 20%    | 38.20 ± 1.3| 37.94 ± 0.6| **45.73 ± 0.4**|
> |               | 30%    | 44.93 ± 0.6| 44.72 ± 0.5| **51.75 ± 0.4**|
> |               | 50%    | 53.81 ± 0.2| 53.37 ± 0.4| **57.88 ± 0.2**|
> | CIFAR 100     | 10%    | 36.68 ± 0.6| 37.65 ± 0.8| **47.58 ± 0.5**|
> |               | 20%    | 53.16 ± 2.2| 52.79 ± 0.8| **61.44 ± 0.7**|
> |               | 30%    | 63.02 ± 1.2| 62.28 ± 1.2| **67.89 ± 0.2**|
> |               | 50%    | 70.68 ± 0.4| 71.19 ± 0.3| **74.03 ± 0.3**|
> | CIFAR 10      | 10%    | 72.26 ± 0.5| 72.65 ± 0.9| **79.47 ± 0.5**|
> |               | 20%    | 84.30 ± 0.9| 84.30 ± 1.2| **87.82 ± 0.9**|
> |               | 30%    | 88.47 ± 0.6| 88.37 ± 1.2| **92.15 ± 0.6**|
> |               | 50%    | 91.89 ± 0.4| 92.67 ± 0.5| **94.36 ± 0.2**|
> | SVHN          | 8%     | 84.31 ± 1.8| 82.31 ± 2.6| **89.52 ± 0.8**|
> |               | 12%    | 88.99 ± 1.0| 88.41 ± 1.3| **93.18 ± 0.5**|
> |               | 16%    | 90.42 ± 0.8| 90.34 ± 0.8| **94.31 ± 0.3**|
> |               | 20%    | 91.56 ± 0.4| 91.95 ± 0.8| **95.08 ± 0.3**|
>
> **Q4: the 'balanced' aspect of the proposed loss; experiments on class imbalanced data**
>
> We would like to clarify that the 'balanced' keyword in our work refers to the interaction of the *input similarity* component and the *label variability* component of the loss bound as derived in Theorem 1 with respect to the subset. Specifically, focusing on improving one component might make another component worse thus there is a need for balancing between those two components.
>
> In our evaluation, we follow the standard setting of most existing works, where the methods are primarily evaluated on balanced data. However, we understand that in a real-life scenario, the data could be imbalanced. So we can certainly evaluate our method in some class imbalance setting.
>
> We have employed two ways of creating imbalanced data: exponential and step-wise. In exponential, we decrease the class size using exponential decay $N_{c_i}\times e^{-0.01i}$ where $N_{c_{i}}$ is the number of samples for class $c_i$. In step-wise imbalance, we prune 80\% of data from 20\% of classes. For the baselines, we chose the most recent competitive methods (CCS and Moderate) in addition to the random method. As can be seen, the results show our method consistently outperforms the competitive baselines under different class imbalance settings.
>
> **To be continued...**

---

> ### Author Response · Authors · 2023-11-18
>
> **Continuing...**
>
> Imbalance result for CIFAR 100, exponential
> | Subset | Random | Moderate | CCS | BOSS |
> |--------|--------|----------|-----|------|
> | 10%    | 27.39 ± 0.9| 25.37 ± 1.7| 29.41 ± 0.5| **36.63 ± 0.9**|
> | 20%    | 42.82 ± 1.1| 40.57 ± 0.6| 44.36 ± 1.7| **50.91 ± 0.6**|
> | 30%    | 52.51 ± 0.7| 50.00 ± 3.0| 50.87 ± 1.4| **57.15 ± 0.5**|
> | 50%    | 63.03 ± 0.6| 61.76 ± 0.2| 61.86 ± 0.5| **66.72 ± 0.1**|
>
> Imbalance result for CIFAR 100, step-wise
> | Subset | Random | Moderate | CCS | BOSS |
> |--------|--------|----------|-----|------|
> | 10%    | 31.66 ± 0.7| 27.02 ± 0.7| 33.60 ± 0.9| **38.72 ± 0.9**|
> | 20%    | 47.36 ± 0.9| 41.77 ± 2.9| 46.82 ± 0.6| **53.75 ± 0.3**|
> | 30%    | 56.64 ± 0.2| 52.31 ± 0.6| 52.81 ± 1.1| **58.37 ± 0.4**|
> | 50%    | 62.19 ± 0.9| 59.96 ± 0.7| 60.25 ± 0.2| **66.03 ± 0.2**|

---

> ### Author Response · Authors · 2023-11-21
>
> We sincerely appreciate your effort in reviewing our paper and the valuable feedback that you have provided. We hope that we have thoroughly addressed your inquiries and cleared any confusion regarding our work. We took into account your suggestion to incorporate additional baselines, conducted an evaluation of our work on an imbalanced dataset, and have accordingly made updates to the main paper. We hope that these responses align with your expectations, and we would be grateful if you would consider revisiting your assessment. We are happy to address any further concerns or queries you may have.

---

### Author Response · Authors · 2023-11-18
**General Response**

In this general response, we would like to highlight the novel contributions of this paper that are relevant to the questions from the reviewers, including:

- proposing a novel loss bound (for **Reviewer jBvt: Q1** and **Reviewer AoWq: Q1**),

- connecting label variability with EL2N score (for **Reviewer AoWq: Q2**),

- connecting subset size and label variability with the importance function (for **Reviewer AoWq: Q3**), and

- proposing a novel objective to jointly maximize input similarity and sample importance (for **Reviewer jBvt: Q3**, **Reviewer BNoZ: Q6**, and **Reviewer AoWq: Q1**).

(We have incorporated the suggested changes in the revised paper with the changes highlighted in brick red.)

**The loss bound:**

First, we propose a novel loss bound concerning optimal core-set selection. Although we also draw inspiration from the classic core-set active learning bound, the balancing of the two components: *input similarity* and *label variability*, is novel and non-trivial. It is an important result derived from our theoretical analysis. In particular, the involvement of the label variability makes our method substantially different from the standard facility location function (**Reviewer jBvt: Q1**). As we show in Appendix D.1, it is not feasible to optimize the objectives individually. The challenge of balancing the two components motivates our selection method (to be expanded later) which goes beyond a simple heuristic (**Reviewer AoWq: Q1**).

In Appendix D.1 of the revised paper, we show a detailed analysis of the feature similarity and label variability components, which demonstrates the difficulty of balancing them (Figures 7 through 9). Let's consider two cases: (1) subset budget is low (Figure 10) and (2) subset budget is high (Figure 12).

 - For *case 1*, selecting representative samples or selecting the major clusters is beneficial because it gives the best input similarity. In contrast, selecting difficult samples with high label variability misses those major clusters making the first component much worse.

 - For *case 2*, we have enough samples to cover the major clusters allowing more points near the boundary to be selected. This minimizes both components while learning the decision boundary better compared to the low-budget case.

**Connecting label variability with EL2N score**

We would like to clarify that our goal is not to directly minimize $||{\bf y}_i-{\bf y}_j||$ (or its bound) individually. Instead, we aim to find a subset that can minimize both the feature distance (first term of Equation 1) and the label variability (second term of Equation 1) between the selected subset and the full set. Theorem 2 shows that the label variability of a data sample to its neighbor is lower bounded by its EL2N score. Thus, a sample with a high EL2N score guarantees to have a large label variability to its neighbor sample (which implies that this sample cannot be represented by its neighbor). Including these samples into the subset will decrease the label variability between the subset and the full set (otherwise, since their labels cannot be represented by the samples in the subset, they will bring a large increase to the overall label variability). As a result, selecting these samples into the subset will make the second term (full set sum of $\lambda^{y}$$||{\bf y}_i-{\bf y}_j||$ ) small. Similarly, choosing the subset of most representative samples will make the first term (full set sum of $\lambda^{\eta}$$||{\bf x}_i-{\bf x}_j||$) small.

**Connecting subset size and label variability with the importance function:**

In order to incorporate the dynamic relation to the subset size, we derive the importance function (Equation 4) as a function of subset size and sample difficulty where the sample difficulty (EL2N) approximates the label variability. Here, the parameter $c$ can be roughly chosen linearly according to the subset size.

**Novel objective:**

This allows us to represent the loss minimization of Equation 1 as joint maximization of input similarity and importance function. To this end, this objective seamlessly integrates all the components: input similarity, label variability, and subset size. The submodular nature of this function also enables efficient subset selection using a greedy algorithm.

---

### Meta-Review · Area_Chair_1fMR · 2023-12-15

**Metareview:**

The paper introduces Balanced One-shot Subset Selection (BOSS), a method for efficient deep-learning training. BOSS selects a subset of data that balances diversity and difficulty, addressing the limitations of existing methods that focus on only one aspect. This approach is grounded in a theoretical loss-bound analysis, ensuring the subset represents the joint data distribution effectively. BOSS's performance, validated through extensive experiments on various datasets, surpasses current methods by optimally balancing these two key factors, influenced by subset size.

The reviewers raised several issues about this work, including the lack of theoretical motivation for the proposed approach, and the heuristic nature of the approach, and the lack of comparison to several important subset selection approaches including repeated random sampling (sampling with replacement subsets of the dataset every time, i.e., a different random subset is selected every time). I agree with the reviewers on this, and I would encourage the authors to provide some theoretical justification as to why an approach like this makes sense from a principled perspective.

**Justification For Why Not Higher Score:**

I think this paper needs some more work before it is ready for acceptance (see all the reviews below).

**Justification For Why Not Lower Score:**

N/A

---

### Decision · Program_Chairs · 2024-01-16

Reject